# Information dynamics and Memory in Neural Networks through Fisher Information Diffusion

**Haodong Qin** [1 2]  **Tatyana O. Sharpee** [1 2]

## Abstract

We present a general theoretical framework for analyzing how information about past inputs is encoded in recurrent networks into evolving dynamics rather than being represented as convergence to static attractors. Using dynamic mean-field theory and diffusion from physics, we derive a Fisher information diffusion operator that links network connectivity structure to the time-resolved propagation of information across interacting subpopulations. The analysis reveals that operating near criticality (spectral radius near one) is necessary but not sufficient for reliable memory in structured or non-normal recurrent networks; effective information retention requires alignment between input–output structure and stable dynamical subspaces. The theory yields principled initialization rules that balance stability and sensitivity, mitigating vanishing and exploding gradients. Experiments on the copy task and sequential MNIST show faster convergence and higher accuracy than standard random initialization. Together, these results provide both principled design guidelines for recurrent networks and new theoretical insight into how information can be preserved over time in their dynamics.

## 1. Introduction

Recurrent neural networks (RNNs) are fundamental models for processing sequential data, and their dynamics have been a central focus in machine learning. At their core, RNNs implement a simple yet powerful principle: they compress an arbitrarily long input sequence into a continuously evolving latent state whose dynamics support prediction, control, and sequence understanding. Central to this capability is the network's ability to retain information about past inputs over time (Sompolinsky et al., 1988; White et al., 2004; Spaak et al., 2017). Understanding how such information is encoded in evolving representations, and how recurrent connectivity governs its retention and transformation, has therefore been a long-standing goal in machine learning.

Traditional theories of memory in recurrent networks emphasize attractor-based associative mechanisms, such as Hopfield networks (Hopfield, 1982; Ramsauer et al., 2021). In these models, inputs are mapped to stationary states, and memory is identified with convergence to fixed points, largely ignoring sensitivity to small perturbations of the input. However, many sequential tasks require a different form of memory: information must remain available while representations continue to evolve, so that small perturbations in earlier inputs remain distinguishable at later times. This motivates a dynamical notion of memory based on sensitivity to past inputs, rather than convergence to static states.

From a dynamical-systems perspective, memory is closely tied to stability. Operating near the transition between stable dynamics and chaos-the edge of chaos-can prolong sensitivity to past inputs and thereby enhance memory (Sompolinsky et al., 1988; Bertschinger et al., 2004; Schuecker et al., 2018). In parallel, optimization-centric studies have emphasized that maintaining stable signal propagation through time is essential for trainability. Early exact analyses of deep linear networks showed how learning dynamics depend strongly on network structure and mode-wise signal propagation (Saxe et al., 2014). Analyses based on Jacobian statistics and dynamical isometry connect gradient stability to the spectrum of recurrent transformations, motivating structured parameterizations and initializations such as identity, orthogonal, or unitary matrices (Le et al., 2015; Henaff et al., 2017; Wisdom et al., 2016; Arjovsky et al., 2016; Pennington et al., 2017).

Yet, stability criteria based on eigenvalues, Lyapunov exponents, or asymptotic behavior are fundamentally coarse. They do not directly quantify how much information about a particular past input is retained at intermediate times, nor how retention is distributed across interacting modules. This gap is especially consequential in non-normal recurrent net-

[1]Department of physics, University of California San Diego, San Diego, USA [2]Salk Institute, San Diego, USA. Correspondence to: Tatyana O. Sharpee <sharpee@salk.edu>.

*Proceedings of the 43rd International Conference on Machine Learning*, Seoul, South Korea. PMLR 306, 2026. Copyright 2026 by the author(s).

works, where stable eigenvalues can coexist with strong transient mixing that redirects sensitivity away from real input–output directions (Christodoulou et al., 2022). More broadly, modern recurrent architectures and biological circuits are inherently modular, comprising multiple interacting subpopulations (e.g., layers, modules, or cell types), where memory depends not only on global stability but also on how information is routed between populations.

Fisher information provides a principled local measure of memory as sensitivity. Rather than evaluating memory via discrete decoding accuracy, Fisher information quantifies the infinitesimal distinguishability of nearby inputs after propagation through the network, and is therefore directly linked to preservation of local input geometry (Ganguli & Sompolinsky, 2010a; Pakrooh et al., 2013). In recurrent systems, Fisher information admits a natural interpretation as a memory curve: in linear RNNs, the Fisher memory matrix characterizes how sensitivity to past inputs decays over time (Ganguli et al., 2008). Fisher-based analyses have also motivated structured and non-normal initializations that enhance memory and trainability in specific architectures (Orhan & Pitkow, 2020).

Despite these advances, existing analytic treatments of memory remain limited in three key ways. First, most Fisher-information and mean-field analyses assume statistically homogeneous connectivity (i.i.d. weights) (Toyoizumi & Abbott, 2011), effectively reducing the network to a single population and yielding memory curves that do not resolve how information is routed across interacting modules. Second, multi-population dynamical mean-field theory (DMFT) has primarily been used to characterize dynamical regimes such as stability and the onset of chaos (Kadmon & Sompolinsky, 2015; Kuśmierz et al., 2025), but it does not provide a closed-form description of time-resolved information retention at the population level. Third, Jacobian-based approaches quantify signal and gradient stability through spectral criteria and related spectral structure in deep networks (Pennington et al., 2018), but they do not yield a tractable, population-resolved theory that predicts where memory is preserved, how it flows between subpopulations, or how connectivity affects memory retention over finite horizons. As a result, it remains unclear how to derive principled design rules that directly connect block-structured connectivity statistics to memory trajectories and memory capacity in modular recurrent networks.

**Contribution.** We introduce a theoretical framework that characterizes memory in modular recurrent networks by deriving a population-resolved, time-resolved description of how information about past inputs is preserved and transmitted through recurrent dynamics. Our approach combines multi-population DMFT with diffusion ideas from physics, enabling analytic predictions beyond the homo-geneous single-population setting. Specifically, we model an RNN as a collection of interacting subpopulations with block-structured random connectivity and nonlinear saturating activation functions (Fig. 1b), and we derive a closed-form evolution equation for population-level sensitivity statistics that quantify memory.

Our central result is an analytic Fisher information diffusion operator: a positive linear operator that governs how Fisher information about past inputs propagates across subpopulations and evolves over time (Fig. 1a). Crucially, this operator provides a direct and constructive link from connectivity statistics to memory dynamics: given only (i) the block-variance structure of recurrent weights, (ii) the activity-dependent nonlinear gain statistics along trajectories, and (iii) the input-output configuration, the theory predicts population-wise Fisher memory curves and enables direct optimization of information retention without training. This diffusion perspective makes two key points explicit:

1. **Criticality is necessary but not sufficient.** While operating near $\rho(A) \approx 1$ prevents information from vanishing asymptotically, effective memory additionally requires that the stable information-carrying subspace is aligned with the input–output structure, particularly in non-normal networks where transient mixing can suppress usable memory (Christodoulou et al., 2022).

2. **Memory is distributed and time-resolved.** The diffusion operator provides an explicit population-wise accounting of how much information is retained at each time step, enabling analytic memory trajectories and capacity predictions in block-structured networks.

Because the Fisher diffusion operator yields an explicit mapping from connectivity statistics to memory retention, it enables principled design rules. We derive Fisher-optimal initialization schemes that balance stability and sensitivity, mitigating vanishing and exploding gradients in structured recurrent networks. We validate the theory by comparing analytic predictions to direct numerical estimates of Fisher information dynamics across subpopulations, and demonstrate improved optimization stability and performance on sequential benchmarks (copy task and sequential MNIST) relative to standard initializations such as Xavier, Kaiming, orthogonal, and unitary across multiple seeds and datasets.

Overall, our contributions are threefold:

1. **Closed-form memory theory for block-structured RNNs.** We derive analytic expressions for time-resolved memory trajectories and capacity in multi-population recurrent networks, extending Fisher-memory analyses beyond the single-population setting.

2. **Fisher information diffusion.** We introduce the Fisher

diffusion operator as a tractable and interpretable tool for quantifying information transmission across interacting subpopulations over time.

3. **Connectivity-to-initialization design rules.** We derive Fisher-optimal initializations directly from population-level connectivity statistics and show consistent empirical benefits on sequential learning tasks.

## 2. Recurrent networks with block-structured connectivity

We begin with the general dynamics of recurrent networks, then introduce a block-structured generalization that enables analysis at the subpopulation level.

**Recurrent dynamics.** We consider a recurrent network of $N$ neurons with discrete-time dynamics

$$h_i(t) = \sum_{j=1}^{N} J_{ij} S_j(t) + \eta_i(t),$$
$$S_j(t) = \phi[w_j x(t) + h_j(t-1)]. \quad (1)$$

where $h_i(t)$ is the internal state of neuron $i$, $S_i(t)$ its output, and $J_{ij}$ is the connectivity matrix connecting the neuron $j$ to neuron $i$. The activation function is $\phi(x) = \tanh(x)$ with $\phi'(0) = 1$. The network receives an external input $x(t)$ through weights $w_i$, and each neuron is driven by independent Gaussian noise $\eta_i(t)$ with zero mean and covariance $\langle \eta_i(t)\eta_j(s)\rangle = \sigma^2 \delta_{ij}\delta_{ts}$.

**Subpopulation structure.** Classical analyses typically assume that the connectivity matrix $J$ is i.i.d. Gaussian (Toyoizumi & Abbott, 2011), corresponding to a single homogeneous population. To capture more general network structures, we partition the network state $\mathbf{h}(t) \in \mathbb{R}^N$ into $M$ subpopulations (Fig. 1b). Each neuron $i$ is assigned a label $m(i) \in \{1, \dots, M\}$, with subpopulation $m$ containing a fraction $f_m$ of the neurons such that $\sum_{m=1}^{M} f_m = 1$. The resulting connectivity matrix $J$ acquires a block structure, where each block encodes connections between two subpopulations (Fig. 1b). This formulation generalizes standard feedforward or layered networks: purely feedforward connectivity appears as a special case, while feedback and skip connections are naturally represented by off-diagonal blocks.

**Block-structured connectivity.** Within the mean-field approximation, weights are modeled as independent Gaussians with zero mean and block-dependent variances: $\langle J_{ij}\rangle_J = 0, \langle J_{ij}^2\rangle_J = \frac{1}{N} g_{m(i)n(j)}^2$. Here $m(i)$ and $n(j)$ denote the subpopulations of neurons $i$ and $j$. Each block encodes

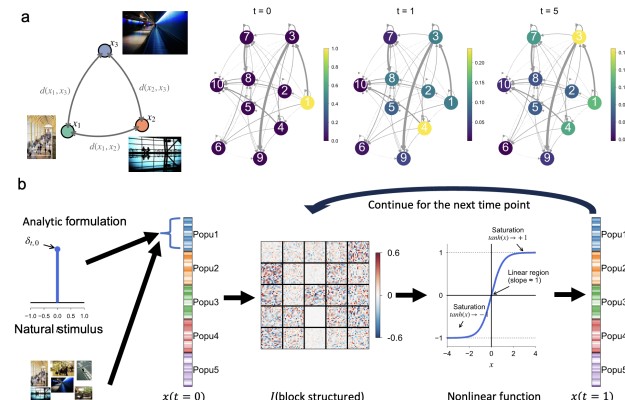

*Figure 1.* **Illustration of Fisher information diffusion and experimental setup.** **(a)** Unlike traditional graph diffusion models, which track the spread of neural activities, we study the diffusion of Fisher information, quantifying how well each subpopulation retains the geometry of the data (i.e., pairwise sample distances $(d(x_i, x_j))$ where $x_i, x_j$ are input samples) over time in a recurrent multi-subpopulation network. **(b)** Schematic of the network architecture and its time evolution. The network consists of multiple subpopulations with recurrent connections drawn from a zero-mean Gaussian distribution with specified variance, representing connection strength. Inputs can be flexibly configured; for clarity, we show the case where either an impulse (for analytic derivation) or natural images (for testing) are provided to the first subpopulation.

connections from subpopulation $n$ to $m$, with variance parameter $g_{mn}^2$ controlling its strength. We will refer to the block-gain matrix $G$ with entries $G_{mn} \equiv g_{mn}^2 f_n$.

**Fisher information.** We probe memory with an impulse input $x(t) = \theta\,\delta_{t,0}$ and study how well its amplitude $\theta$ is preserved across time and populations. The Fisher information (FI) about $\theta$ is

$$\mathcal{I}(\theta, t) = -\mathbb{E}_{p(\mathbf{h}(t)|\theta)}\left[\frac{\partial^2}{\partial\theta^2}\log p(\mathbf{h}(t) \mid \theta)\right], \quad (2)$$

which defines the Fisher memory curve (Ganguli et al., 2008). In the dynamic mean-field (DMFT) limit $N \to \infty$ with block-structured random connectivity $J$, the population-level pre-activations become self-averaging and are well-approximated by an effective Gaussian process, as in classical DMFT analyses (Ganguli et al., 2008; Toyoizumi & Abbott, 2011). In particular, the mean-field covariance concentrates to a diagonal form,

$$\langle \Sigma_{ij}(t)\rangle_J = \delta_{ij}\, q_{m(j)}, \quad (3)$$

where $q_m$ denotes the stationary pre-activation variance within subpopulation $m$. Under DMFT, the Fisher information admits a closed-form expression (Appendix A.2.6):

$$\mathcal{I}(\theta, t) = N \sum_m \frac{f_m}{q_m} \left\langle \left( \frac{\partial \mu_m(t)}{\partial \theta} \right)^2 \right\rangle_J,$$

$$q_m = \sigma^2 + \sum_n G_{mn} \langle S_n^2 \rangle_n. \tag{4}$$

where $\mu_m(t) \equiv \langle h_m(t) \mid J \rangle$. Although $\mu_m(t) = 0$ on average, FI remains nonzero since it depends on the variance of the sensitivity $\partial \mu_m(t) / \partial \theta$, which is shaped by both the nonlinearity $\phi$ and the inter-population connectivity. Importantly, this sensitivity is not constant over time and differs across populations.

**Fisher information diffusion.** We have derived the Fisher diffusion operator $A$ that propagates sensitivities $\left\langle (\partial \mu_m(t)/\partial \theta)^2 \right\rangle_J$ across subpopulations from one time step to the next (derivation in Appendix A.2). For two subpopulations, it factorizes into a connectivity and a sensitivity term:

$$A = \underbrace{\begin{pmatrix} G_{11} & G_{12} \\ G_{21} & G_{22} \end{pmatrix}}_{\text{Connectivity}} \cdot \underbrace{\begin{pmatrix} \langle (S')^2 \rangle_1 & 0 \\ 0 & \langle (S')^2 \rangle_2 \end{pmatrix}}_{\text{Sensitivity/Gain}}, \tag{5}$$

where $\langle (S')^2 \rangle_n$ denotes the mean squared derivative of the activation function in subpopulation $n$. The connectivity block captures how information is routed between groups, while the sensitivity block captures how nonlinear responses modulate this transfer. In Appendix A.2 we provide an analytic expression for $\langle (S')^2 \rangle_n$ that depends solely on the block-gain matrix $G$, making the operator fully determined by network connectivity structure. In the linear limit—when the activation function is purely linear—one has $\langle (S')^2 \rangle_n = 1$ for all subpopulations. In this case the sensitivity block reduces to the identity, and the Fisher information diffusion operator $A$ coincides with the block-gain matrix $G$ itself.

Repeated application of $A$ describes how information flows across populations. The total FI at time $t$ is then $\frac{\mathcal{I}(t)}{N} = \sum_{mn} \frac{f_m}{q_m} (A^t)_{mn} w_n^2$, and $\mathcal{I}(0) = 0$. Crucially, although individual neural activities evolve nonlinearly, the collective statistics of subpopulations can be expressed as evolving linearly under the diffusion operator. The nonlinearity is absorbed into the term $\langle (S')^2 \rangle_n$, which is a nonlinear function of the block gains $G$. In this way, the operator acts analogously to a transfer matrix in graph diffusion, providing a linear structure that enables a tractable analytic description of how connectivity shapes the encoding and preservation of information over time. We present two tests to evaluate our analytic characterization of Fisher information.

**Direct Fisher quantification.** First, we directly estimate Fisher information from network simulations. We simulate a network with $N = 10{,}000$ neurons, $f_1 = f_2 = 0.5$,

$\sigma = 0.1$, and input weights $w_1 = 1, w_2 = 0$, so that the impulse is applied only to the first subpopulation. For input amplitudes $\theta \in \{0, -0.1, 0.1\}$, we compute the sensitivity term $\left\langle (\partial \mu_m(t)/\partial \theta)^2 \right\rangle_J$ in Eq. (4) (see Appendix A.3). To test how connectivity motifs affect information dynamics, we consider three configurations: (i) self-recurrence only, (ii) feedforward coupling, and (iii) feedback coupling (Fig. 2a–c). Across all cases, the analytic solution based on the diffusion operator accurately matches simulations—capturing not only the magnitude of Fisher information in each population, but also the temporal dynamics, including oscillatory flow between subpopulations. Finally, we examine how the agreement scales with network size $N$: the mean-squared error between simulated and analytic trajectories decreases rapidly and becomes negligible for $N \geq 1000$ (Fig. 9).

**Preservation of input geometry.** We next tested whether Fisher information predicts how well a network preserves the geometry of natural inputs—defined as the pairwise distances between input stimuli (Ganguli & Sompolinsky, 2010a; Pakrooh et al., 2013). Below, we provide an intuitive argument showing that Fisher-optimal connectivity conditions coincide with those required for local isometry preservation.

**Proposition 2.1** (Connecting Fisher information with preservation of geometry)**.** *Consider a recurrent network with block-gain matrix $G$ and activation function $\phi$. Under the mean-field approximation, optimal information retention—defined as preservation of local geometry between stimulus representations—is achieved when $G \langle \phi'^2 \rangle = 1$. This is precisely the Fisher-information criterion for non-vanishing memory.*

*Sketch proof.* The result follows by connecting ideas from compressed sensing and nonlinear mean-field theory.

*(1) Linear case and RIP.* For a linear map $f(x) = Jx$, the Restricted Isometry Property (RIP) (Foucart & Rauhut, 2013) ensures approximate distance preservation: $\|f(u) - f(v)\|^2 \approx \|u - v\|^2$. In particular, if $J_{ij} \sim \mathcal{N}(0, g^2/N)$, then $J^\top J$ concentrates around $g^2 I$ in the large-$N$ limit (up to random fluctuations). Thus choosing the critical scaling $g^2 = 1$ makes $J$ approximately norm-preserving, yielding near-isometry (and hence RIP on low-dimensional or sparse subsets with high probability (Foucart & Rauhut, 2013)).

*(2) Nonlinear extension.* With nonlinearity $\phi$, local distances transform as $\|f(x) - f(x')\|^2 \approx \|\phi'(x) J(x - x')\|^2$. Replacing $\phi'(x)^2$ by its population average under mean-field theory gives the effective gain condition $G \langle \phi'^2 \rangle \approx 1$.

*(3) Fisher information connection.* From the Fisher diffusion framework, sustained (non-decaying) memory requires that the leading eigenvalue of $G\langle \phi'^2 \rangle$ equals 1.

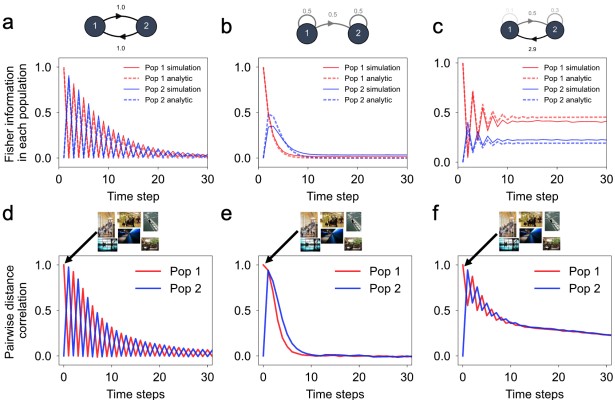

*Figure 2.* **Empirical validation of the Fisher diffusion framework. (a–c)** Time evolution of Fisher information in two subpopulations after an input pulse to Population 1 at $t = 0$, under three motifs: (a) self-recurrence only, (b) feedforward only, and (c) a recurrent architecture close to the analytic optimum. Solid lines show simulation results; dashed lines show analytic predictions from the FI diffusion operator. The analytic framework accurately captures both the magnitude and temporal dynamics, including oscillatory information flow. **(d–f)** Geometry preservation for the same networks for IndoorCVPR_09 images. Pairwise correlations between input distances and network representations quantify how well each architecture maintains input geometry. The results confirm that architectures predicted to optimize Fisher retention also preserve stimulus geometry more effectively.

Thus, the conditions for (i) preserving local geometry (via RIP/Johnson–Lindenstrauss (Foucart & Rauhut, 2013)) and (ii) sustaining Fisher information are identical. □

**Empirical validation on real images.** We presented 15,619 IndoorCVPR_09 images (flattened to dimension 7500) as inputs to the first subpopulation of a network with $N = 15{,}000$, $f_1 = f_2 = 0.5$, and $\sigma = 0.1$. Each image was processed individually, and at time $t$ we recorded the activity vectors of both subpopulations as the network's internal representations. To quantify geometry preservation, we computed all pairwise Euclidean distances between the original images, and likewise all pairwise distances between their corresponding neural representations (see Appendix A.4). We then measured the correlation between these two distance matrices: a correlation of 1 would indicate perfect isometry (geometry preserved exactly), while lower correlations reflect increasing distortion. This correlation therefore serves as a direct measure of how faithfully the network preserves the relational structure of its inputs (Fig. 2d–f). Although this metric is distinct from Fisher information, it produces the same qualitative conclusions: the analytic framework accurately predicts both the oscillatory dynamics of information flow and the relative ability of different network motifs to preserve input geometry.

This empirical test highlights a key difference from Hopfield networks. In Hopfield models, memory is implemented by fixed-point attractors, and capacity is limited by the number of such stable states that can be stored. In our framework, by contrast, memory is defined by how well the differences between stimuli are preserved as activity evolves. This capacity does not depend on the number of stimuli presented and dataset indepent, but instead on whether the network size $N$ is sufficiently large relative to the sparsity of the input space—a condition closely analogous to the Restricted Isometry Property (RIP) for Gaussian matrices(Foucart & Rauhut, 2013). We verify this by evaluating the same networks (Fig. 2) on CIFAR-10 images and observe highly consistent information dynamics across datasets. In particular, the trajectories of Fisher information dynamics are strongly correlated between CIFAR-10 and IndoorCVPR_09 (Pearson correlations $0.993, 0.992, 0.980$

In contrast to classical dynamical-systems analyses of network stability, which are fundamentally asymptotic and only characterize whether information decays to zero in the long-time limit, the Fisher information diffusion operator provides a time-resolved, population-specific quantification of information retention, revealing the intermediate-time dynamics of information storage and transfer that are not captured by stability criteria alone.

## 3. Conditions for Optimal Fisher Information

In the context of information diffusion, achieving maximal long-term retention of an input requires two conditions on the diffusion operator $A$:

1. **Criticality.** The spectral radius of $A$ must satisfy $\rho(A) = \max_i |\lambda_i| = 1$. If $\rho(A) < 1$, Fisher information decays exponentially; if $\rho(A) > 1$, it diverges uncontrollably. Criticality therefore guarantees the dynamic stability (Kadmon & Sompolinsky, 2015) such that information does not vanish at long times, but on its own it is not sufficient for optimal retention.

2. **Eigenvector alignment and transient information.** Let $v$ denote the normalized right eigenvector associated with the leading eigenvalue $\lambda_{\max} = 1$. Asymptotically, $\lim_{t \to \infty} \mathcal{I}(t) \propto \|(\mathbf{w}^\top v)v\|_1$, where $\mathbf{w}$ is the input configuration. Thus, only the input component aligned with $v$ is retained. A complication is that $A$ is generally non-normal (not symmetric), so its eigenvalues and eigenvectors may be complex. Since the input configuration $\mathbf{w}$ is real, the effective alignment with complex eigenvectors can be small, limiting long-term retention. In practice, explicitly computing eigenvalues and eigenvectors is costly for networks with many subpopulations. By contrast, computing Fisher information over time only requires iterated multiplication by $A$, which is more scalable. Moreover, by choosing the integration horizon $T$, one can tune the emphasis between transient retention

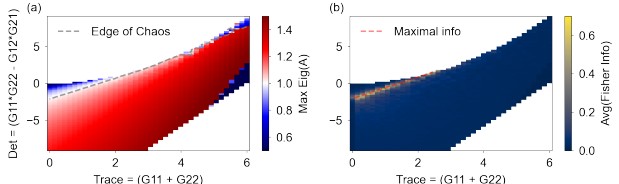

*Figure 3.* **Edge of chaos is necessary but not sufficient for optimal Fisher information retention.** (a) Phase diagram of the spectral radius $\rho(A)$ as a function of intrinsic network parameters, parameterized by $\mathrm{Tr}(G)$ and $\det(G)$, obtained from the analytic solution of the Fisher information diffusion operator (Eq. 23). The gray dashed curve indicates the critical boundary $\rho(A) = 1$. (b) Average Fisher information per time step over 100 steps, showing that maximal information retention occurs only within a restricted band of $\mathrm{Tr}(G)$, despite stability near the critical boundary.

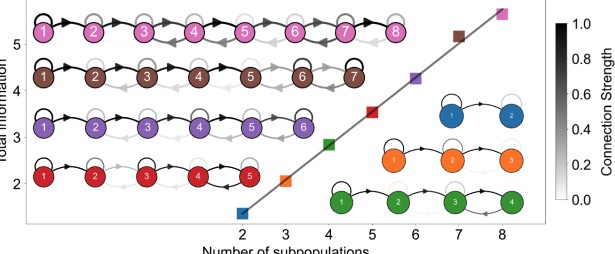

*Figure 4.* **Connectivity structures and information retention across subpopulations.** Inset diagrams show optimized connectivity patterns for networks with 2–8 subpopulations, revealing strong feedforward pathways, moderate self-recurrence, and sparse feedback loops. The main panel demonstrates that total Fisher information retention scales linearly with the number of subpopulations, indicating that greater modular depth enhances memory capacity.

and long-term stability. A practical design objective is therefore to maximize the average Fisher information over a finite horizon: $\overline{\mathcal{I}} = \frac{1}{T} \sum_{t=1}^{T} \mathcal{I}(t)$. This metric balances stability at criticality with the preservation of transient information.

To illustrate these conditions, we analyze a simple two-population recurrent network. The block gain matrix is $G = \begin{pmatrix} G_{11} & G_{12} \\ G_{21} & G_{22} \end{pmatrix}$, so that the Fisher diffusion operator $A$ depends on the four parameters $G_{mn}$. Because $A$ can be summarized in terms of its trace and determinant, the parameter space can be reduced from four to two dimensions, enabling a clear visualization. We perform a dense grid search over all $G_{mn} \in [0,3]$. For each parameter setting, we compute: The spectral radius $\rho(A)$, used to identify the critical boundary $\rho(A) = 1$ (Fig. 3a), and the average Fisher information $\overline{\mathcal{I}}$ across 100 timesteps, aggregated over both populations (Fig. 3b). We observe that:

1. The critical boundary (grey dashed line) extends across the full range of $\mathrm{Tr}(G)$, confirming that criticality is a necessary condition for sustained information flow.

2. Optimal information retention occurs only along the critical boundary but is restricted to a narrower band of $\mathrm{Tr}(G)$, showing that criticality alone is not sufficient.

This demonstrates that while criticality is required for sustained information flow, alignment of the stable diffusion direction $v$ with the input configuration $\mathbf{w}$ is additionally necessary to achieve optimal Fisher information retention.

## 4. Optimal Structure for Longer Chains

Our analytical framework extends naturally to networks of many subpopulations or deeper recurrent structures. For clarity and tractability, we focus on sequential chains

in which only adjacent subpopulations are connected on through adjustable self-recurrent, feedforward, and feedback links (Fig. 4). The connectivity is captured by a generalized Toeplitz-like gain matrix $G$ where only $G_{mm}$, $G_{m,m+1}$, and $G_{m+1,m}$ are nonzero, preserving the chain structure while allowing parameter flexibility. All subpopulations are equal in size ($f_m = 1/M$), and input is applied only to the first subpopulation ($w_i = \delta_{i,1}$).

Within this architecture we build the Fisher information functional in terms of the allowed block gains $G_{mn}$ and maximize the time- and population-averaged Fisher information $\overline{\mathcal{I}}$ over $T = 100$ using differential evolution. The optimization reveals clear design principles: strong feedforward connections propagate signals efficiently, while carefully placed feedback stabilizes and modulates this flow. Indiscriminate feedback is detrimental; instead, optimal networks exhibit sparse, strategically positioned feedback links that break the chain into nested loops for robust information retention. The characteristic broken-feedback pattern can be intuitively justified in the linear limit (Appendix A.5).

Finally, we find a striking scaling law (Fig. 4): keeping the total number of neurons fixed—yet large enough for mean-field theory—the network's total Fisher information grows approximately linearly with the number of subpopulations. Thus, deeper or more finely partitioned chains intrinsically possess greater information capacity when their connectivity is properly optimized.

## 5. Sequential stimulus

For an optimal network, neural activity evolves in an information-preserving regime, rather than rapidly collapsing to a fixed point that erases sensitivity to past inputs. When an input is projected into the network, the internal representation changes continuously over time instead of remaining static. As a result, identical stimuli injected at dif-

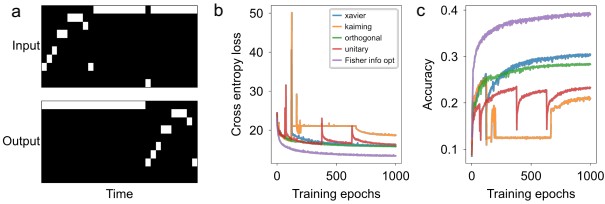

*Figure 5.* **Sequential memory test on the copy task.** (a) Schematic of the copy task. (b) Training performance for a representative setting with delay length $T_{\text{delay}} = 50$ and random seed $= 10$. Recurrent networks initialized using Fisher-information-optimized weights converge faster and achieve higher accuracy than networks with standard random initialization. Results for additional delay lengths and random seeds are shown in Figs. 12, 13, and 14.

ferent times can lead to distinct downstream representations at later time. This continual drift of stimulus representations provides a natural mechanism for encoding the order of sequential inputs, akin to the function of working memory.

Once a block-gain matrix $G$ optimal for Fisher information is found for a given input configuration, a corresponding block-structured connectivity matrix $J$ can be readily constructed. Specifically, we sample the elements of $J$ from zero-mean Gaussians with variances derived from the corresponding entries of $G$. This simple procedure initializes a recurrent network to operate near the edge of chaos, providing a principled starting point for training. Importantly, the sub-population partition is not used to optimize the architecture of the RNN. Rather, it serves as a compact parameterization for constructing principled initializations.

We evaluate whether Fisher-information–optimal initialization improves learning on sequential tasks using two complementary settings. First, we test end-to-end training on the copy task, where all network parameters (including recurrent weights) are trainable. Second, motivated by the hypothesis that improved performance arises from more efficient information propagation from the input subpopulation to the readout subpopulation for decoding, we perform a controlled experiment on sequential MNIST where we fix the recurrent weights and train only the readout. This isolates the contribution of initialization-induced dynamics from that of recurrent weight learning.

Our experiments are designed to isolate the effect of initialization rather than architectural improvements. We therefore follow the standard simple nonlinear RNN setup used in sequential memory benchmarks such as the copy task and sequential MNIST (Keller et al., 2024).

**Copy task** The copy task follows the standard setup of (Keller et al., 2024). Each input sequence has length $T_{\text{delay}} + 20$. The first 10 tokens are random one-hot vectors in categories $\{1, \ldots, 8\}$, followed by $T_{\text{delay}}$ zeros, a single

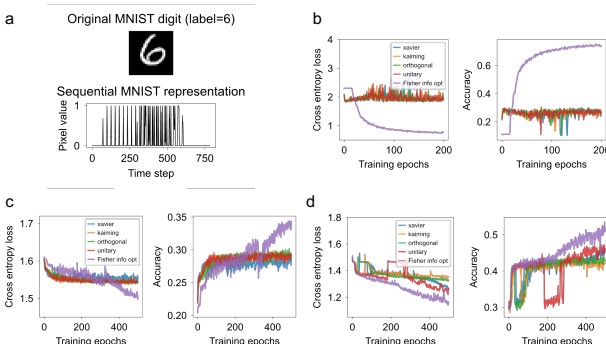

*Figure 6.* **Sequential memory on Sequential MNIST.** (a) In the Sequential MNIST task, each MNIST image is flattened and presented as a sequence of 784 pixel values, one pixel per time step. (b) Recurrent networks initialized with Fisher-information-optimized weights converge substantially faster during training (left) and achieve higher final classification accuracy (right) compared to networks with standard random initialization. We further evaluate the approach on more challenging sequential image classification tasks, including CIFAR-10 (c) and IndoorCVPR_09 (d). Results across multiple random seeds and additional experimental details for (c) and (d) are provided in Appendix A.6.3.

delimiter token (category 9), and finally 9 more zeros. The target output has the same length but remains zero until the final 10 steps, where it reproduces the initial random sequence (Fig. 5a). This task probes a network's ability to encode categorical information and maintain it in memory for $T_{\text{delay}}$ time steps before recall.

We trained a simple RNN of 100 neurons with a `tanh` nonlinearity and $T_{\text{delay}} = 50$. For Fisher-information-optimized initialization, the network is partitioned into 10 equal subpopulations (matching the input dimension), with purely feedforward connections $G_{m+1,m} = 1 \, (m = 1, \ldots, 9)$ and a single feedback link $G_{1,10} = 1$. The corresponding Fisher diffusion matrix $A$ thus has spectral radius one. Weights from input to RNN are sent such that input stimulus are passed to the first subpopulation.

Across a range of delay lengths and random seeds, Fisher information optimal initialization consistently yields faster convergence and higher final accuracy than widely used baselines, including Xavier, Kaiming, orthogonal, and unitary initializations (learning rate $= 10^{-3}$; A representative example with $T_{\text{delay}} = 50$ and random seed $= 10$ is shown in Fig. 5b, with complete results reported in Figs. 12, 13, and 14. These results support the hypothesis that Fisher-optimal initial conditions provide a favorable dynamical regime in which information about early inputs can propagate through the recurrent dynamics efficiently, reducing the burden and time on optimization to discover such regimes from scratch.

**Sequential MNIST** In sequential MNIST, the 784 pixels of each image are presented one at a time, and classification

is performed from the final hidden state (Fig. 6a). To directly test whether the improved learning observed in the copy task arises from more effective information propagation induced by the initialization, we perform a controlled experiment in which the recurrent weights are fixed after initialization and only the readout is trained. We use the same RNN as in the copy task, and implement the readout as a one-layer MLP with tanh nonlinearity, which improves performance in practice.

Under this setting, the network dynamics throughout training are entirely determined by the initialization of the recurrent weights. We find that Fisher-information–optimal initialization achieves both faster convergence and higher final accuracy than Xavier, Kaiming, orthogonal, and unitary baselines (Fig. 6b). This supports the interpretation that Fisher information optimal connectivity promotes stable yet sensitive dynamics that preserve information about early inputs over long horizons, thereby improving downstream decoding even when the recurrent core is not trained.

Finally, to assess dataset-independence, we extend the same fixed-recurrence protocol to CIFAR-10 and IndoorCVPR_09 (Fig. 6c-d). Results across multiple random seeds are shown in Figs. 17 and 18, with additional experimental details provided in Appendix A.6.3. Across datasets and random seeds (Appendix A.6.3), Fisher-information–optimal networks consistently converge faster and achieve higher accuracy than existing initialization schemes. Together, these results suggest that the primary benefit of Fisher-information–optimal initialization is dynamical rather than task-specific: it places the recurrent network in a stable yet sensitive regime in which input perturbations propagate reliably across subpopulations, thereby keeping task-relevant information accessible to the readout throughout training.

# 6. Discussion

Our contributions are: (i) We introduced a block-structured, mean-field framework in which the Fisher diffusion operator analytically tracks how information flows across interacting subpopulations in recurrent networks. (ii) Criticality (spectral radius $\approx 1$) is necessary but not sufficient for long-term retention, and alignment between input structure and the stable subspace is equally essential. (iii) From these principles we derived simple, Fisher-information–optimized initializations that (empirically) accelerate training and improve accuracy on sequential memory tasks.

The analytic expression linking connectivity structure to Fisher-information optimality yields a principled, theory-driven initialization rule for RNNs. A key advantage is that a large recurrent weight matrix can be configured by only a small set of population-level block gains $g_{ij}$, each determining the variance of the Gaussian weights within a connectiv-

ity block. This mapping from a few block-level parameters to the full recurrent matrix places the network directly in the Fisher information optimal regime, where sensitivity to perturbations of past inputs is preserved. Such initialization is beneficial for sequential memory tasks, in which late-time activity must remain sensitive to inputs presented many steps earlier. Fisher information quantifies sensitivity: classical fixed-point models such as Hopfield networks intentionally collapse perturbations to enforce convergence and therefore cannot maintain the fine-grained distinctions required for dynamic memory. In contrast, standard initialization schemes are not memory-aware and place the network in a generic region of the high-dimensional parameter space, forcing optimization to discover configurations that preserve long-range sensitivity—a process that is unstable and prone to regions where gradients vanish or explode. By initializing connectivity directly in the Fisher-optimal regime, our framework avoids these difficulties and provides a theoretically grounded method for stabilizing information flow in RNNs from the outset.

Although the main goal here is to extract Fisher infomraiton in a nonlinear RNN. In the linear limit—when the activation function is purely linear—one has $\langle (S')^2 \rangle_n = 1$ for all subpopulations. In this case the sensitivity block reduces to the identity, and the Fisher information diffusion operator $A$ coincides with the block-gain matrix $G$ itself, which can also used for multi-layer linear RNN or State space models.

Our framework recovers linear recurrent networks as a special case. In the linear limit $\phi(x) = x$, the Fisher information diffusion operator reduces to the block-gain matrix $G$. The analysis therefore directly applies to multi-layer linear RNNs and discrete-time linear state-space models, including modern structured state-space sequence models (Gu et al., 2022; Gu & Dao, 2024). This reduction serves as a consistency check and clarifies that the novelty of the framework lies in its extension to nonlinear, modular recurrent networks, where information transport becomes activity-dependent.

Our analysis focuses on single-state recurrent networks, where the Fisher–information dynamics can be characterized analytically. Architectures such as LSTMs and GRUs incorporate multiple coupled states and multiplicative gates, placing them outside the formal scope of the theory. Nevertheless, because each gate contains a recurrent transformation structurally similar to the dynamics we analyze, we evaluated whether Fisher–optimal initialization might still offer practical benefits at the gate level. As discussed in Appendix A.6.2, we find that applying the proposed initialization to these recurrent components improves training performance on sequential memory tasks, suggesting that the underlying principles may extend beyond the analytically tractable setting.

Beyond machine learning, this framework offers an abstract theoretical lens for thinking about working and short-term memory in biological systems. By explicitly characterizing how sensitivity to past inputs can be distributed and maintained across interacting neural populations, our results complement existing neuroscience perspectives that emphasize dynamic, population-level representations of memory. We discuss these conceptual connections in the related work section (Appendix A.1).

## Acknowledgements

We thank Arjun Karuvally for the inspiring suggestion to connect our theoretical framework on Fisher information and sequential memory to practical machine learning applications, particularly network initialization. We also thank Terrence Sejnowski and Arjun Karuvally for insightful conversations on traveling waves and working memory, which helped open up a direction we had not previously considered.

## Impact Statement

This paper presents work whose goal is to advance the field of Machine Learning. There are many potential societal consequences of our work, none which we feel must be specifically highlighted here.

## Conflict of Interest Disclosure.

The authors declare that they have no financial conflicts of interest related to this work.

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

# A. Appendix

## A.1. Related work

### Distributed population dynamics and working-memory codes

There is now a rich literature showing that memory is implemented by distributed, population-level dynamics rather than single "memory cells." For example, Cavanagh et al. (2018); Spaak et al. (2017) analyze how working-memory representations in primate prefrontal cortex can be supported by combinations of persistent and dynamic population codes. More precisely, Spaak et al. (2017) show that during a memory-guided saccade task, despite dynamic population coding, the representational geometry of working memory remains stable. Meyers et al. (2008) demonstrate dynamic population coding of category information in ITC and PFC during delay tasks, emphasizing time-varying trajectories rather than static tuning. Finally, Kurtkaya et al. (2025) and other recent RNN studies characterize dynamical "phases" (such as limit cycles, slow manifolds, sequential activity) that support short-term memory in trained recurrent networks.

Our contribution is complementary to this line of work. The biological studies above collectively demonstrate that neural populations encode task-relevant information in the geometry of their activity trajectories, and that this information can persist even when individual neurons exhibit only transient or heterogeneous responses. Building on this assumption, we develop a closed-form theoretical framework that *quantifies* how much information about input differences can be preserved by such population dynamics. Rather than analyzing decoding accuracy from recorded neural activity, we derive Fisher-information–based expressions that make explicit how information retention depends on population-level connectivity motifs—including the trace and determinant of the connectivity matrix. In other words, whereas prior biological and computational work establishes *that* distributed neural trajectories carry short-term and working-memory information, our framework provides an analytic characterization of *how much* of the input geometry can be preserved, and *under which dynamical regimes*, directly from the parameters governing the network's connectivity structure.

### Fisher information in machine learning and dynamical networks

Prior work has used Fisher information (FI) to analyze memory and information retention in recurrent neural dynamics, though almost entirely in homogeneous or unstructured settings. Ganguli et al. (2008) and later Ganguli & Sompolinsky (2010b) applied FI to characterize how recurrent linear systems maintain short-term memory of past inputs and established fundamental limits on memory capacity. A parallel body of work in computational neuroscience has focused on linear Fisher information, which provides a closed-form expression for stimulus discriminability in recurrently connected populations (Beck et al., 2011) and has been widely used experimentally to estimate information from correlated population activity (Kanitscheider et al., 2015).

Beyond recurrent dynamics, Fisher information has a long history in machine learning as a metric that shapes optimization, curvature, and gradient propagation. Amari et al. (2019) analyses natural-gradient learning in random deep networks; Pennington & Worah (2018) characterize the eigenvalue spectrum of the Fisher Information Matrix (FIM) in wide networks; Karakida et al. (2019; 2020) study universal statistics and pathological curvature regimes induced by extreme FIM anisotropy; and Karakida & Osawa (2020) show that approximate natural-gradient methods still inherit fast-convergence guarantees in wide limits. Hayase & Karakida (2021) further demonstrate that networks satisfying dynamical isometry nevertheless develop concentrated FIM spectra that require depth-dependent learning-rate scaling, directly linking FI to vanishing and exploding gradients.

Relative to these lines of work, our contribution is different in scope and objective. Prior studies either focused on homogeneous single-population RNNs, linearized dynamics, or the role of Fisher information in optimization geometry. By contrast, our framework derives explicit, analytic conditions under which *multi-population* recurrent networks preserve Fisher information about early inputs over time, thereby maintaining the geometry of input differences along dynamically evolving trajectories.

Crucially, these conditions provide a direct link between the *connectivity structure* of the network—specifically, population-level parameters such as the trace and determinant of the $G$ matrix—and the amount of Fisher information that can persist along recurrent dynamics. This connection has practical consequences for sequential-memory learning. Instead of relying on gradient descent to discover a narrow region of parameter space that supports long-range sensitivity to inputs, our Fisher-optimal initialization places the network in a regime that already retains information about distant past stimuli. Because large RNNs have high-dimensional, non-convex loss landscapes, standard initializations that are not memory-

optimized often struggle to reach these regions and can become trapped in poor local minima where gradients vanish. In contrast, Fisher-optimal initialization tunes the block-structured gains so that a large number of individual recurrent weights are automatically set to a configuration that preserves input geometry from the outset, thereby accelerating training and improving final performance.

### A.2. Analytic Fisher Information for Multiple Sub-populations

We consider a recurrent network divided into multiple subpopulations. For each sub-population $m$, let

$$\mu_m(t) = \langle h_m(t) \,|\, J \rangle. \tag{6}$$

denote the mean activity at time $t$, averaged over dynamic noise but at fixed synaptic matrix $J$. Our goal is to compute the Fisher information $\mathcal{I}_m(t)$ of $\mu_m(t)$ with respect to an input parameter $\theta$:

$$\mathcal{I}(\theta, t) = K \sum_m \frac{f_m}{q_m} \left\langle \left( \frac{\partial \mu_m(t)}{\partial \theta} \right)^2 \right\rangle_J. \tag{7}$$

The key is to derive the analytic formula for $\left\langle \left( \frac{\partial \mu_m(t)}{\partial \theta} \right)^2 \right\rangle_J$.

#### A.2.1. REPLICA TRICK FOR THE DERIVATIVE OF THE MEAN

Introducing replicas $a, b$, we have

$$\left\langle \left( \frac{\partial \mu_m(t)}{\partial \theta} \right)^2 \right\rangle_J = \left\langle \left[ \frac{\partial}{\partial \theta} \langle h_m(t)|J \rangle \right]^2 \right\rangle_J = \left\langle \frac{\partial}{\partial \theta^a} \langle h_m^a(t)|J \rangle \frac{\partial}{\partial \theta^b} \langle h_m^b(t)|J \rangle \right\rangle_J$$

$$= \frac{\partial^2}{\partial \theta^a \partial \theta^b} \langle h_m^a(t) h_m^b(t) \rangle = \frac{\partial^2}{\partial \theta^a \partial \theta^b} q_m^{ab}(t), \tag{8}$$

where $q_m^{ab}(t) \equiv \langle h_m^a(t) \, h_m^b(t) \rangle_m$.

#### A.2.2. MEAN-FIELD EXPRESSION FOR THE CORRELATION

In mean-field, the correlation splits into an i.i.d. noise term and a term generated by recurrent inputs:

$$q_m^{ab}(t, s) = \sigma^2 \delta_{ab} \delta_{ts} + \sum_n g_{mn}^2 f_n \langle S_n^a(t) S_n^b(s) \rangle_n = \sigma^2 + \sum_n G_{mn} C_n^{ab}(t, s), \tag{9}$$

with $G_{mn} \equiv g_{mn}^2 f_n$ and $C_n^{ab}(t, s) = \langle S_n^a(t) S_n^b(s) \rangle$, the firing rate correlation in sub-population $n$.

#### A.2.3. DIFFERENTIATING FIRING RATE CORRELATION FUNCTION

Applying $\partial_\theta^a \partial_\theta^b$ to $C_n^{ab}(t, s)$ and using the chain rule for the nonlinearity $\phi(\cdot)$ (with $\phi' = \frac{d\phi}{dz}$) yields

$$\begin{aligned}
\partial^{ab} C_n^{ab}(t) &= \partial^{ab} \langle \phi^a(t) \phi^b(t) \rangle_n \\
&= \partial^{ab} \langle \phi(w_n \Theta^a(t-1) + x^a(t-1)) \phi(w_n \Theta^b(t-1) + x^b(t-1)) \rangle_n \\
&= \langle \phi'^a \cdot (w_n + \partial_a x^a(t-1)) \, \phi'^b \cdot (w_n + \partial_b x^b(t-1)) \rangle_n \\
&= \langle \phi'^a \phi'^b \rangle_n \langle (w_n^2 + (\partial_a x^a(t-1) + \partial_b x^b(t-1)) w_n + \partial_a x^a(t-1) \partial_b x^b(t-1) \rangle_n \\
&= \langle \phi'^a \phi'^b \rangle_n (w_n^2 + \partial_a \partial_b \langle x^a(t-1) x^b(t-1) \rangle_n \\
&= \langle \phi'^a \phi'^b \rangle_n (w_n^2 + \partial_a \partial_b \, q_n^{ab}(t-1)).
\end{aligned} \tag{10}$$

The first order in line 4 with terms $\langle \partial_a x^a \rangle_n = \partial_a \langle x^a \rangle_n = \partial_a 0 = 0$.

#### A.2.4. RECURRENCE FOR THE SECOND DERIVATIVE OF POPULATION SPECIFIC VARIANCE

Combining (9) and (10) produces a linear recurrence:

$$\partial^a \partial^b q_m^{ab}(t+1) = \sum_n G_{mn} \langle \phi'^a \phi'^b \rangle_n (\partial^a \partial^b q_n^{ab}(t) + w_n^2 \delta_{t,0}), \qquad A_{mn} \equiv G_{mn} \langle \phi'^a \phi'^b \rangle_n. \tag{11}$$

Because $q^{ab}$ depends only on earlier inputs, the initial condition is $\partial_\theta^a \partial_\theta^b q_n^{ab}(t) = 0$ for $t \leq 0$. Iterating (11) once at $t = 0$ gives

$$\partial_\theta^a \partial_\theta^b q_m^{ab}(1) \;=\; \sum_{n=1}^{M} A_{mn} w_n^2, \tag{12}$$

Since the network only receives the input at $t = 0$, each successive iteration amounts to matrix multiplication by $A_{mn} \equiv G_{mn} \langle \phi'^a \phi'^b \rangle_n$. Repeating the recurrence $t$ times results:

$$\partial^a \partial^b q_{m,t+1}^{ab} = \sum_n (\mathbf{A}^{t+1})_{mn} w_n^2 \,. \tag{13}$$

### A.2.5. CLOSED-FORM FISHER INFORMATION

Finally, substituting Eq. (13) into (8) gives

$$\left\langle \left( \frac{\partial \mu_m(t)}{\partial \theta} \right)^2 \right\rangle_J = \sum_n (\mathbf{A}^t)_{mn} w_n^2, \quad A_{mn} \equiv G_{mn} \langle (\phi')^2 \rangle_n \,. \tag{14}$$

Eq. (14) shows that the propagation of the Fisher information through the network can be effectively captured by the Fisher information diffusion operator $A$.

### A.2.6. ANALYTIC DERIVATION OF THE FISHER INFORMATION DIFFUSION OPERATOR

To obtain an analytical expression for the Fisher information in relation to the optimal connectivity parameters $G_{mn}$ and to gain an intuitive understanding of Fisher information from a network perspective, two key tasks are essential:

1. Analytically resolve the self-consistent equations for $q_1$ and $q_2$ to understand the dynamics in the system. These solutions also allow us to construct Gaussian probability distributions with variances $q_1$ and $q_2$, respectively. From these distributions, we compute the second-order moments $\langle S^2 \rangle$ and $\langle (S')^2 \rangle$, which are essential for determining the Fisher information diffusion operator.

2. Derive an analytical formula for Fisher information that elucidates the relationship between network characteristics, the connection between populations of neurons, and optimal information capacity.

### A.2.7. ANALYTIC CALCULATION OF THE MEAN SQUARED HYPERBOLIC TANGENT

Since both the self consistent equations

$$\begin{aligned} q_1 &= \sigma_1^2 = \sigma^2 + G_{11} \langle S^2 \rangle_1 + G_{12} \langle S^2 \rangle_2 \,, \\ q_2 &= \sigma_2^2 = \sigma^2 + G_{22} \langle S^2 \rangle_2 + G_{21} \langle S^2 \rangle_1 \,. \end{aligned} \tag{15}$$

and the Fisher information diffusion operator $A_{mn} = G_{mn} \langle (S')^2 \rangle \rangle_n$. With $S = \tanh(x)$, $S^2$ and $S'$ are highly nonlinear and non local, the values are not close to 0 or 1. As a result, using Taylor expansion of the $\tanh(x)$ produces both poor approximation and analytic challenge when calculating the Gaussian average $\langle S^2 \rangle$ and $\langle (S')^2 \rangle$. We notice that both expressions $S^2$ and $S'$ only relate to the some form of Gaussian average $\langle \tanh^2(x) \rangle$, and we can approximate the $\tanh^2(x)$ with $1 - \exp \frac{x^2}{2\epsilon^2}$ (See Fig. 7 ). Note that this expression insures $(1 - \exp \frac{x^2}{2\epsilon^2})|_{x=0} = 0$. The optimal parameter $\epsilon = 0.7784$ can be derived from the minimizing the integral difference $\int_{-\infty}^{\infty} |1 - \exp \frac{x^2}{2\epsilon^2} - \tanh^2(x)|$. The Gaussian average $\langle f(x) \rangle_\sigma = \int_{-\infty}^{\infty} f(x) N(0, \sigma^2)$ can be calculated easily with a simple form:

$$\begin{aligned} \langle \tanh^2(x) \rangle_i &= 1 - \frac{1}{\sqrt{1 + (\frac{\sigma_i}{\epsilon})^2}} = 1 - \frac{1}{\sqrt{1 + \mu_i^2}}, \mu_i = \frac{\sigma_i}{\epsilon} \,, \\ \langle \tanh'(x)^2 \rangle_i &= \langle (1 - \tanh^2(x))^2 \rangle_i = \frac{1}{\sqrt{1 + 2\mu_i^2}} \,. \end{aligned} \tag{16}$$

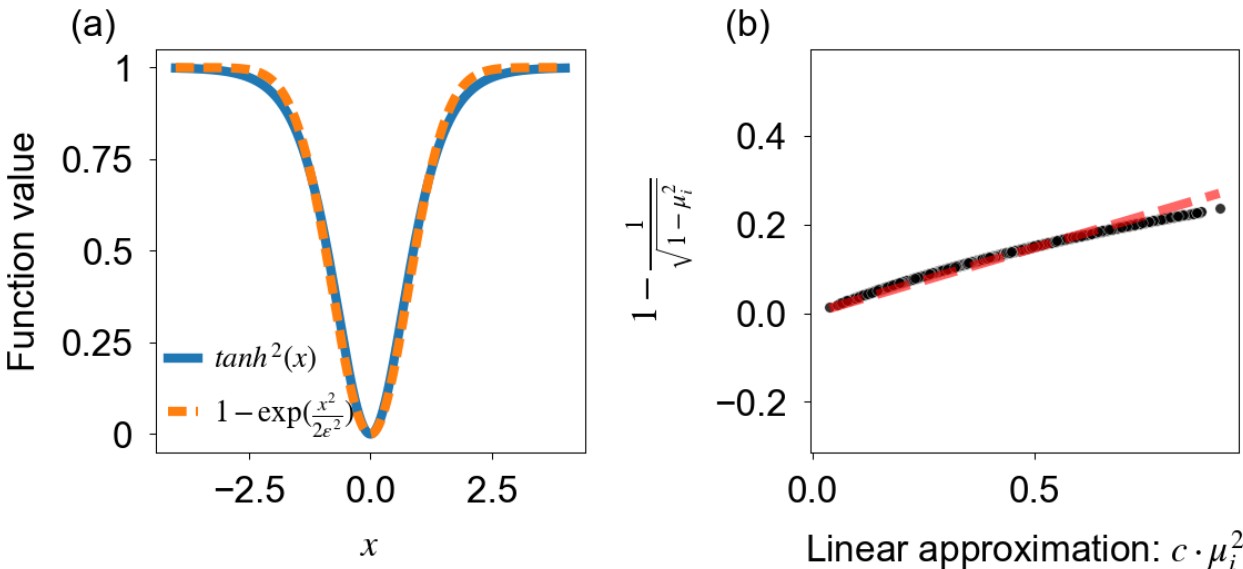

*Figure 7.* (a) Approximation of the nonlinear function $\tanh^2(x)$ using the surrogate form $f(x, \epsilon) = 1 - \exp(-x^2/2\epsilon^2)$, with the optimal parameter $\epsilon = 0.7784$. (b) Linear approximation of the expression $1 - \frac{1}{\sqrt{1+\mu_i^2}}$ using $c \cdot \mu_i^2$, where the optimal slope $c = 0.2948$ is determined via least-squares fitting. Each point $\mu_i$ is obtained from a grid search over network configurations in the two-population case, constrained to operate near the edge of chaos ($|\rho(A) - 1| < 0.1$), where $\rho(A)$ denotes the spectral radius of the effective connectivity matrix.

For numeric solutions, Eq. ((15)) ((20)) extends naturally to an arbitrary number of sub-populations, yield a closed system of nonlinear equations for the variance $q_i$. This system can be efficiently solved using the 'fsolve' function from the 'scipy.optimize' package. Once the $q_i$ are obtained, they are substituted into Eq. ((20)) to evaluate the Fisher-information diffusion operator.

A.2.8. ANALYTIC CALCULATION OF THE POPULATION SPECIFIC ORDER PARAMETERS

With Eq. ((16)), we can rewrite the self consistent equations:

$$\mu_1^2 = \mu^2 + M_{11}(1 - \frac{1}{\sqrt{1+\mu_1^2}}) + M_{12}(1 - \frac{1}{\sqrt{1+\mu_2^2}}) \,,$$
$$\mu_2^2 = \mu^2 + M_{21}(1 - \frac{1}{\sqrt{1+\mu_1^2}}) + M_{22}(1 - \frac{1}{\sqrt{1+\mu_2^2}}) \,, \tag{17}$$
$$M_{mn} = G_{mn}/\epsilon^2, \ \mu_i^2 = \sigma_i^2/\epsilon^2, \ \mu^2 = \sigma^2/\epsilon^2 \,.$$

Solving the Eq. ((17)) directly is difficult and will lead to unintuitive expression since this is a system of cubic equations with non uniform power in each term. For systems operating near the edge of chaos—characterized by a spectral radius close to one ($|\rho(A) - 1| < 0.1$)—the variable $\mu_i$ remains small (Fig 7 ). In this regime, we can approximate the nonlinear expression $1 - \frac{1}{\sqrt{1+\mu_i^2}}$ using a linearized form. Specifically, we use a least-squares fit to determine the optimal slope $c$ in the following approximation:

$$1 - \frac{1}{\sqrt{1+\mu_i^2}} \approx c\mu_i^2, \quad i \in \{1,2\}, \quad c = 0.2948 \,. \tag{18}$$

This approximation simplifies further analysis while preserving accuracy in the small-$\mu_i$ limit.

With the linear approximation Eq. ((18)), the solutions to the self consistent equations are:

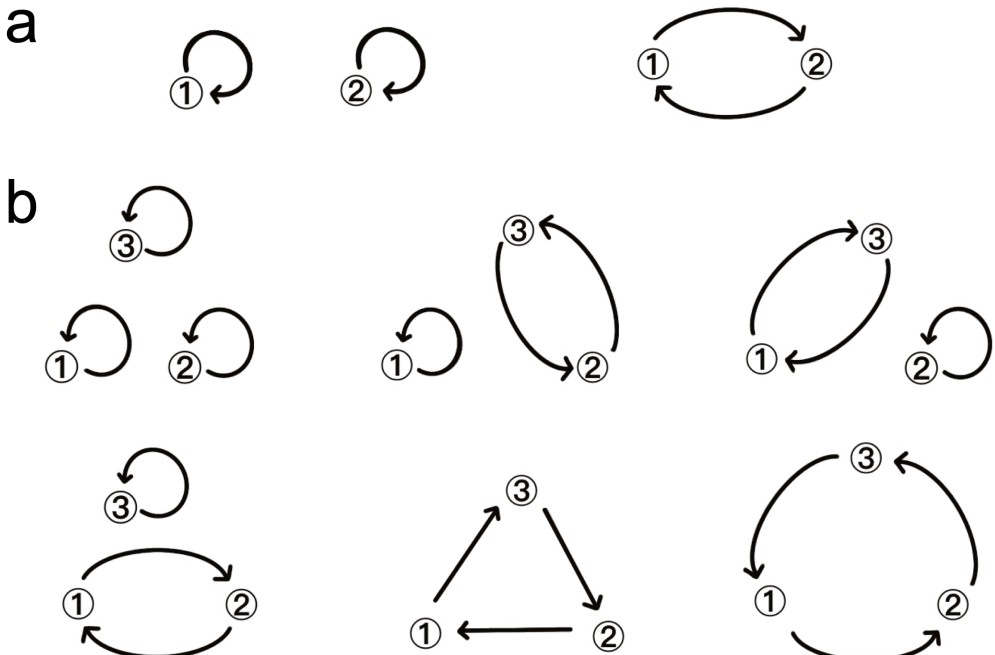

*Figure 8.* A illustration of the determinant of the weight matrix for 2 populations(a) and 3 populations (b).

$$\mu_1^2 \approx \frac{\mu^2 + c\mu^2(M_{12} - M_{22})}{1 - c\,\text{Tr}(M) + c^2\det(M)}\,,$$

$$\mu_2^2 \approx \frac{\mu^2 + c\mu^2(M_{21} - M_{11})}{1 - c\,\text{Tr}(M) + c^2\det(M)}\,,$$

$$\det(M) = M_{11}M_{22} - M_{12}M_{21},\ \text{Tr}(M) = M_{11} + M_{22}\,.$$

(19)

Under the meanfield, subpopulation 1 and subpopulation 2 are considered as nodes with weight matrix as $M_{mn} = G_{mn}/\epsilon^2$. This mapping makes every term in the analytic expression of Eq. ((19)) a familiar graph invariant. The trace, $\text{Tr}(M) = \sum_m M_{mm}$, equals the total weight of all self-loops in the network. Meanwhile, the determinant, $\det(M)$, via the Leibniz expansion, becomes a signed sum over all cycle covers (loop configurations), each monomial corresponding to a distinct set of loops weighted by the product of edge weights (Harary, 1962). These loop configurations are illustrated in Fig. 8.

### A.2.9. ANALYTIC EXPRESSION OF THE FISHER INFORMATION DIFFUSION OPERATOR

The Fisher information diffusion operator $A_{mn} = G_{mn}\langle(S')^2\rangle_n = \epsilon^2 M_{mn}\langle(S')^2\rangle_n$. In Eq. ((19)), we have derived analytic expression in terms of the scaled conductivities $M_{mn} = G_{mn}/\epsilon^2$ and plugging in Eq. ((16)). Directly substitute the Eq. ((19)) into Eq. ((16)), we get:

$$\langle (S')^2 \rangle_1 = \frac{1}{\sqrt{1 + 2\mu_1^2}} \approx f(M)((1 + c(M_{12} - M_{22}))\,,$$

$$\langle (S')^2 \rangle_2 = \frac{1}{\sqrt{1 + 2\mu_2^2}} \approx f(M)((1 + c(M_{21} - M_{11}))\,,$$

$$f(M) = \frac{\mu^2}{1 - c\,\mathrm{Tr}(M) + c^2\,\det(M)}\,, \tag{20}$$

$$\tilde{f}(M) = 1 - 2c\frac{\mu^2}{1 - c\,\mathrm{Tr}(M) + c^2\,\det(M)} = 1 - 2c\,f(M)\,,$$

$$c = 0.2948\,.$$

$$A = \epsilon^2 \begin{pmatrix} M_{11}\langle (S')^2 \rangle_1 & M_{12}\langle (S')^2 \rangle_2 \\ M_{21}\langle (S')^2 \rangle_1 & M_{22}\langle (S')^2 \rangle_2 \end{pmatrix} = \epsilon^2 \begin{pmatrix} M_{11} & M_{12} \\ M_{21} & M_{22} \end{pmatrix} \begin{pmatrix} \langle (S')^2 \rangle_1 & 0 \\ 0 & \langle (S')^2 \rangle_2 \end{pmatrix}\,, \tag{21}$$

$$\epsilon = 0.7784\,.$$

At criticality—i.e. on the "edge of chaos"—the Fisher-information diffusion operator $A$ acquires an eigenvalue exactly equal to unity. Equivalently:

$$\det\big(I - A_{\mathrm{opt}}\big) = 0. \tag{22}$$

By expanding $\det(I - A)$ for our two-population system and grouping terms, we obtain the expression for the condition for the edge of chaos in a fully symmetric form with respect to subpopulations:

$$
\begin{aligned}
0 &= \det(I - A_{opt}) \\
&= 1 - \epsilon^2 (\langle (S')^2 \rangle_1 M_{11} + \langle (S')^2 \rangle_2 M_{22}) + \epsilon^4 \langle (S')^2 \rangle_1 \langle (S')^2 \rangle_2 \det(M) \\
&= 1 - \epsilon^2\,L_1(M) + \epsilon^4\,\det(M)\,L_2(M)\,, \\
L_1(M) &= [\tilde{f}(M) - 2c^2\,f(M)\,\mathrm{Tr}_{\mathrm{off}}(M)]\,\mathrm{Tr}(M) + 2c^2\,f(M)[M_{11}d_2 + M_{22}d_1]\,, \\
L_2(M) &= \tilde{f}^2(M) + 2c^2 f(M)[\mathrm{Tr}(M) - \mathrm{Tr}_{\mathrm{off}}(M)] \\
&\quad + 4c^4 f^2(M)(d_1 d_2 - \mathrm{Tr}(M)\,\mathrm{Tr}_{\mathrm{off}}(M))\,, \\
f(M) &= \frac{\mu^2}{1 - c\,\mathrm{Tr}(M) + c^2\,\det(M)}, \quad \tilde{f}(M) = 1 - 2c\,f(M)\,, \\
d_i &= \sum_k M_{ik} = M_{i1} + M_{i2}, \quad \mathrm{Tr}_{\mathrm{off}}(M) = M_{12} + M_{21},\,, \\
c &= 0.2948,\ \epsilon = 0.7784\,.
\end{aligned}
\tag{23}
$$

where we recognize:

1. **Trace**, $\mathrm{Tr}(M) = M_{11} + M_{22}$.
   The *total self-loop weight* (sum of length-1 cycles), which sets first-order feedback gain.

2. **Off-diagonal trace, $\mathrm{Tr}_{\mathrm{off}}(M) = M_{12} + M_{21}$.**
   The total cross-population coupling, measuring the strength of two-node interactions.

3. **Determinant,** $\det(M) = M_{11}M_{22} - M_{12}M_{21}$.
   A signed sum over all 2-cycle covers:

   - $M_{11}M_{22}$ counts two independent self-loops,
   - $M_{12}M_{21}$ counts the reciprocal 2-node cycle.

The determinant provides insights into the connectivity and spanning trees of a graph, as detailed in the Matrix-Tree Theorem (Harary, 1962).

4. **Weighted in-degrees.**

$$d_i = \sum_k M_{ik} = M_{i1} + M_{i2},$$

the total incoming weight to subpopulation $i$. The concept of in-degree is a basic measure in graph theory, indicating the number of edges arriving at a node (Diestel, 2005).

Here, through algebraic manipulation and careful rearrangement, we derive a form of the edge-of-chaos condition that is symmetric across subpopulations and expressed entirely in terms of familiar graph-theoretic quantities—such as trace, off-diagonal trace, determinant, and in-degree of the connectivity matrix. This reformulation reveals how the topology of structured neural networks directly shapes the onset of criticality.

## A.3. Numeric calculation of Fisher information from Monte-Carlo

In the main text, we benchmark the analytic expression of the Fisher information against a direct Monte-Carlo estimate obtained from explicit simulations of the recurrent neural network (RNN). The numerical procedure consists of three main stages: (i) initialization of the random block-structured connectivity, (ii) simulation of neural trajectories under baseline and perturbed inputs, and (iii) estimation of derivatives via symmetric finite differences.

### A.3.1. NETWORK INITIALIZATION

The network consists of $N$ neurons partitioned into $M$ subpopulations with sizes $n_m = f_m N$. Synaptic connectivity is represented by a block-structured random matrix $J \in \mathbb{R}^{N \times N}$. Each block $J_{mn}$ is sampled i.i.d. from a Gaussian distribution

$$J_{kl} \sim \mathcal{N}\left(0, \ \frac{g_{mn}^2}{N}\right), \qquad k \in \text{pop}_m, \ \ l \in \text{pop}_n,$$

where $g_{mn}$ denotes the population-dependent gain parameter. This choice controls the effective recurrent gain while ensuring that connectivity statistics remain stable as $N$ increases. Here $k$ indexes a postsynaptic neuron in population $m$ and $l$ indexes a presynaptic neuron in population $n$. This blockwise construction ensures that the recurrent connectivity statistics are determined by the gain matrix $g$ while preserving the correct population sizes.

**Dynamical simulation.** Neural activity is described by pre-activations $x_t \in \mathbb{R}^N$ and firing rates $S_t = \tanh(x_t)$. The recurrent dynamics evolve according to

$$x_{t+1} = J S_t + \sigma \, \xi_t, \qquad \xi_t \sim \mathcal{N}(0, I),$$

with additive Gaussian noise of variance $\sigma^2$. At initialization, an external input $\theta$ is injected into the first population, implemented by setting $x_{1:n_1} \leftarrow \theta$. Multiple trajectories are simulated in parallel to estimate ensemble averages.

### A.3.2. PERTURBATION PROTOCOL

To estimate the Fisher information with respect to the input parameter $\theta$, we simulate network dynamics under three input conditions: baseline $\theta$, positively perturbed $\theta + \Delta\theta$, and negatively perturbed $\theta - \Delta\theta$. For each condition, we record the full trajectory of neural activities $\{\mathbf{h}(t)\}_{t=1}^T$. Throughout the simulations, we set the baseline input to $\theta = 0$.

### A.3.3. FISHER INFORMATION ESTIMATION

The sensitivity of mean activity to $\theta$ is approximated via symmetric finite differences:

$$\frac{\partial \mu_t}{\partial \theta} \approx \frac{\mu_t(\theta + \Delta\theta) - \mu_t(\theta - \Delta\theta)}{2\Delta\theta},$$

where $\mu_t(\theta)$ is the average firing rate at time $t$ across trajectories. Squaring and averaging these derivatives over neurons within population $m$ yields a time-resolved Fisher information stored in each subpopulation about the input stimulus over time as in Eq (4)

Here, we also show the MSE between the simulated fisher information and the analytic prediction of the fisher information in Fig. 9

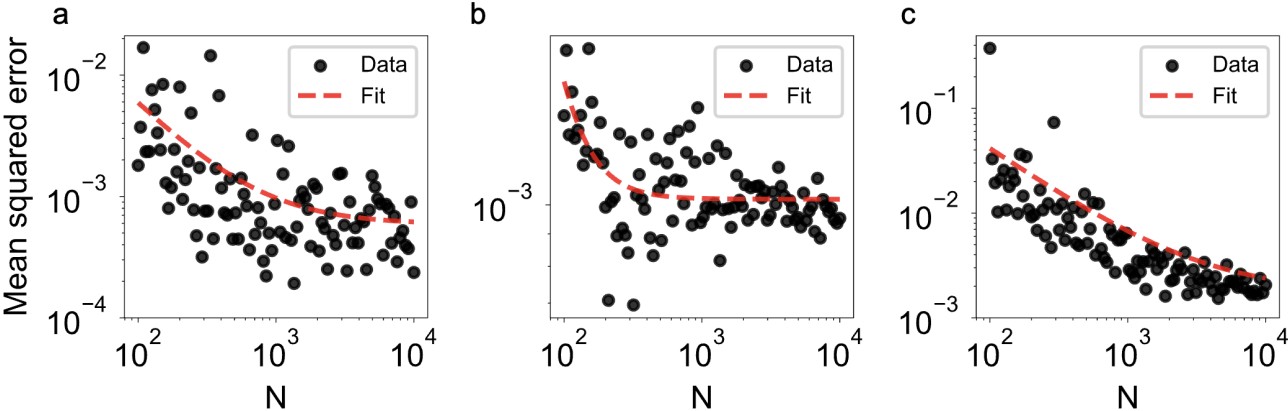

*Figure 9.* **Convergence of simulated Fisher information to analytic predictions.** Mean-squared error (MSE) between Fisher information trajectories obtained from simulations and from the analytic diffusion operator, across the three connectivity configurations in Fig. 2a–c. The MSE decreases rapidly with network size and becomes negligible for $N \geq 1000$. An exponential fit (dashed line) is shown to highlight the convergence trend.

## A.4. Test of Fisher information with natural images as input

**Experimental setup.** To evaluate whether Fisher information predicts geometry preservation, we tested the framework using natural inputs. We used 15,619 IndoorCVPR_09 images, each flattened to a 7,500-dimensional vector, and presented them to the first subpopulation of a two-population recurrent network with $N = 15,000$ neurons, $f_1 = f_2 = 0.5$, and noise variance $\sigma = 0.1$. Each image was processed independently, and at each time step $t$ we recorded the activities of both subpopulations.

**Measuring geometry preservation.** Geometry preservation was quantified by comparing pairwise distances between images in the input space to pairwise distances between their corresponding neural representations. Specifically: 1. For the input set, we computed all pairwise Euclidean distances $D_{\text{input}}(i,j) = \|x_i - x_j\|_2$ between the flattened image vectors. 2. For the network representations at time $t$, we computed analogous pairwise distances $D_{\text{rep}}(i,j) = \|h_i(t) - h_j(t)\|_2$ for each subpopulation. 3. To assess how faithfully the network preserved geometry, we calculated the Pearson correlation coefficient between the upper triangular entries of the two distance matrices,

$$\rho(t) = \text{corr}(\text{vec}(D_{\text{input}}),\ \text{vec}(D_{\text{rep}}(t))),$$

where $\rho = 1$ indicates perfect isometry (exact geometry preservation) and lower values indicate increasing distortion.

This procedure yields a time series $\rho_m(t)$ for each subpopulation $m$, quantifying how input geometry is preserved over time as information diffuses through the network.

**Interpretation.** Although this metric differs from Fisher information, it recovers the same qualitative behavior. In particular, the analytic framework predicts both (i) the oscillatory dynamics of information flow across subpopulations and (ii) the relative ability of different connectivity motifs to preserve the geometry of natural images.

In the main text Theorem 2.1, we provided an intuitive argument—rooted in compressed sensing and the Restricted Isometry Property (RIP)—that a Fisher-optimal (geometrically neutral) initialization approximately preserves pairwise distances for any sufficiently high-dimensional input ensemble, provided the input sparsity remains below the effective dimensionality of the network (Foucart & Rauhut, 2013). In this regime, the encoding dynamics are expected to generalize across datasets and input statistics.

To test this prediction, we repeated the full geometry-preservation analysis using the CIFAR-10 dataset (32×32 RGB images across 10 classes), scaled to the same dimensionality as the IndoorCVPR_09 images (flattened to dimension 7500). The resulting information-preservation dynamics are nearly identical to those shown in Fig.2d–f. Quantitatively, the Pearson correlations between the IndoorCVPR and CIFAR-10 information-flow trajectories are exceptionally high (0.993, 0.992, and 0.980; all $p \ll 10^{-60}$), confirming that the Fisher-optimal initialization induces dataset-independent encoding dynamics.

Our framework views memory as dynamic geometry preservation rather than static attractor storage. This perspective naturally supports generalization at the encoding stage: once initialized at the Fisher-optimal point, the network preserves the relational geometry of novel, unseen inputs without additional training. The consistency of results across IndoorCVPR and CIFAR-10 demonstrates this theoretical prediction—geometry-preserving dynamics arise from the structure of the Fisher-optimized initialization itself, not from dataset-specific learning.

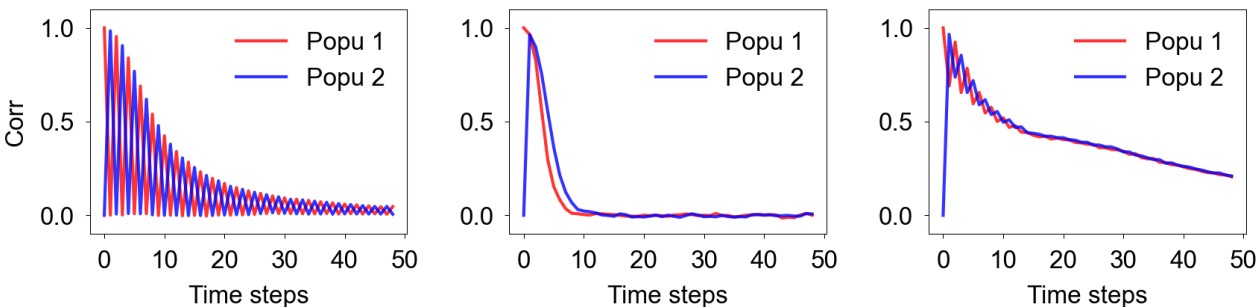

*Figure 10.* **Geometry preservation for CIFAR-10 inputs.** Analysis analogous to Fig. 2, showing pairwise correlations between input distances and network representations. Architectures predicted to optimize Fisher retention also best preserve input geometry. The Pearson correlations between the IndoorCVPR and CIFAR-10 information-flow trajectories (0.993, 0.992, and 0.980; all $p \ll 10^{-60}$) confirm that Fisher-information–optimized initialization preserves pairwise distances independently of the specific input ensemble, provided the input sparsity is below the effective dimensionality of the network.

### A.5. Proof of Optimal Structure for Fisher Information in a Chained Linear Network

In the linear limit—when the activation function is purely linear—one has $\langle (S')^2 \rangle_n = 1$ for all subpopulations. The sensitivity block in the Fisher diffusion operator therefore reduces to the identity, so that

$$A_{\text{linear}} = G.$$

Without loss of generality, consider a chain of four subpopulations with unit self-recurrence:

$$A_{\text{linear}} = G = \begin{pmatrix} 1 & G_{12} & 0 & 0 \\ G_{21} & 1 & G_{23} & 0 \\ 0 & G_{32} & 1 & G_{34} \\ 0 & 0 & G_{43} & 1 \end{pmatrix}. \tag{24}$$

Optimal Fisher information requires that the spectral radius of $A$ equals one, i.e. the largest eigenvalue satisfies $\lambda_{\max} = 1$. Equivalently,

$$0 = \det(A - I) = \det \begin{pmatrix} 0 & G_{12} & 0 & 0 \\ G_{21} & 0 & G_{23} & 0 \\ 0 & G_{32} & 0 & G_{34} \\ 0 & 0 & G_{43} & 0 \end{pmatrix} = G_{12}G_{21}G_{34}G_{43}.$$

For efficient information transmission, the input must enter the first subpopulation and propagate forward through the chain. Hence the forward gains $G_{21}$ and $G_{43}$ cannot vanish. To satisfy $\det(A - I) = 0$, we therefore require

$$G_{12} = 0 \quad \text{or} \quad G_{34} = 0.$$

This condition eliminates the global feedback loops that would otherwise close the chain, giving rise to the broken-loop optimal structure illustrated in Fig. 4. The exact values for the block gain matrix $G$ for each optimized network is shown in Fig. 11

### A.6. Additional experimental test on sequential memory task

#### A.6.1. COPY TASK

**Test on multiple delays.** To further validate our theoretical predictions, we conducted additional sequential-memory experiments across multiple delay lengths ($T_{\text{delay}} = 40, 50, 60$) and multiple random seeds (see Figs. 12, 13, and 14).

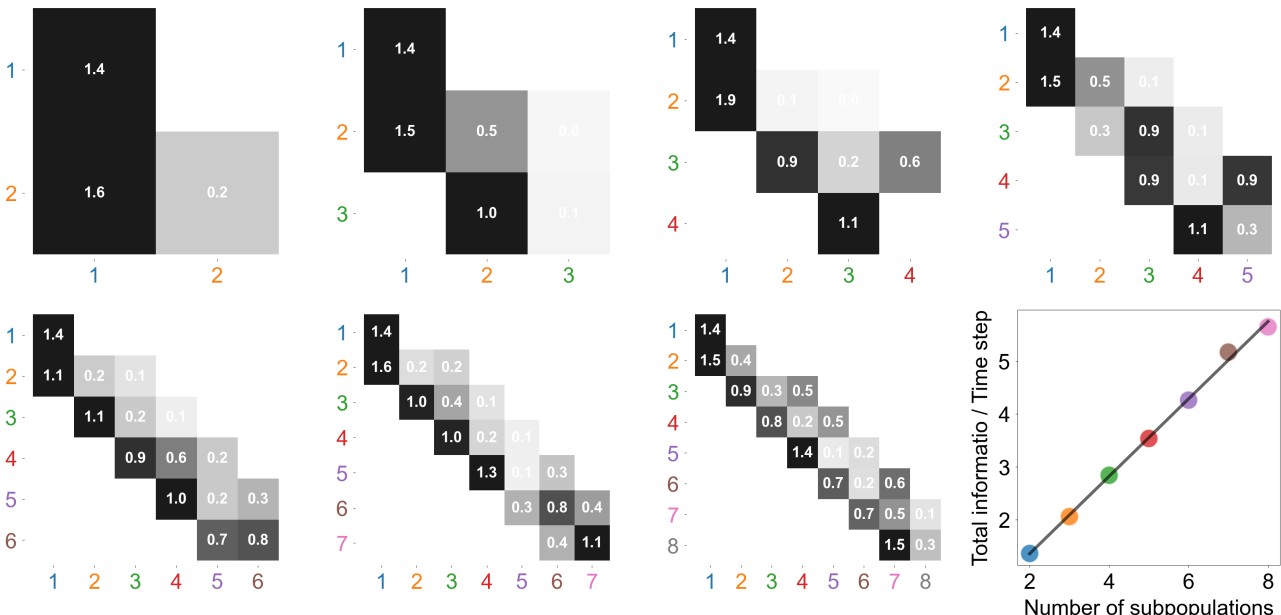

*Figure 11.* Optimized connectivity matrices $G$ for networks composed of 2 to 8 neuronal subpopulations (insets). Each tick color corresponds to a different number of subpopulations. Matrix element $G_{mn}$ represents the strength of the connection from subpopulation $n$ to subpopulation $m$. The final panel illustrates that the total Fisher information $\overline{\mathcal{I}}$ scales approximately linearly with the number of subpopulations in the optimally connected network.

Networks initialized with Fisher–information–optimized weights retain sensitivity to input perturbations occurring far in the past, thereby preserving stimulus information over long times while preventing vanishing or exploding gradients. This leads to more stable dynamics and more efficient training of the decoder.

**Comparison with the common initialization schemes.** We further compared Fisher–optimized initialization with standard schemes, including Xavier, Kaiming, orthogonal, and unitary initialization. Across all delay lengths and seeds, Fisher–information–optimized initialization consistently achieved lower loss, higher accuracy, and faster convergence. These results support the theoretical prediction that operating near the Fisher–optimal regime preserves temporal sensitivity and stabilizes gradient flow, enabling more reliable and efficient learning in sequential-memory tasks.

A.6.2. APPLYING FISHER–OPTIMAL INITIALIZATION TO LSTM GATES.

We further evaluated whether Fisher–information–optimized initialization can provide benefits in more complex recurrent architectures such as LSTMs and GRUs. For clarity, we recall the standard forward pass of an LSTM with forget gate (Hochreiter & Schmidhuber, 1997):

$$
\begin{aligned}
f_t &= \sigma_g(W_f x_t + U_f h_{t-1} + b_f), \\
i_t &= \sigma_g(W_i x_t + U_i h_{t-1} + b_i), \\
o_t &= \sigma_g(W_o x_t + U_o h_{t-1} + b_o), \\
\tilde{c}_t &= \sigma_c(W_c x_t + U_c h_{t-1} + b_c), \\
c_t &= f_t \odot c_{t-1} + i_t \odot \tilde{c}_t, \\
h_t &= o_t \odot \sigma_h(c_t).
\end{aligned}
\tag{25}
$$

Unlike the simple RNN model studied in our theory, LSTMs maintain *two* hidden states ($c_t$ and $h_t$), and the dynamics cannot be written as a single recurrent update of the form

$$
h_t = \phi(W_{ih} x_t + W_{hh} h_{t-1} + b),
\tag{26}
$$

which is the setting under which our Fisher–information analysis is derived. Consequently, our theory does *not* describe the

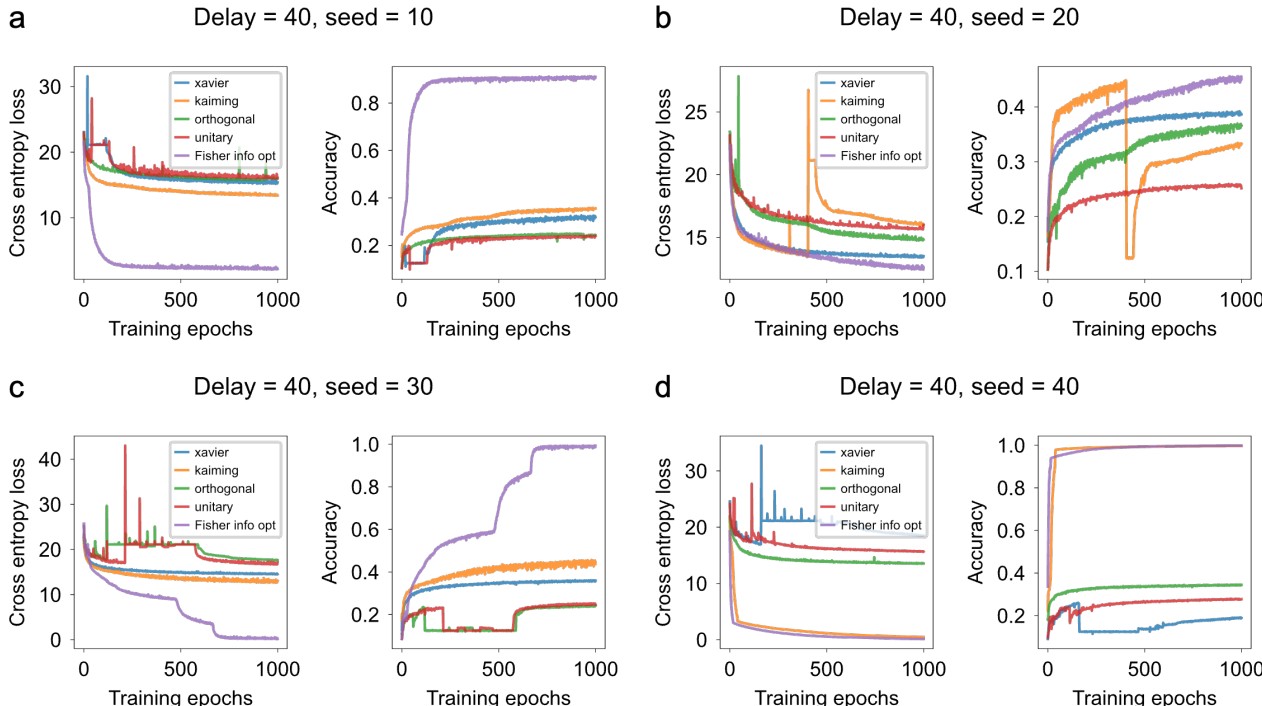

*Figure 12.* **Copy Task with RNN.** Comparison of Fisher–information–optimized initialization with standard initialization schemes (Xavier, Kaiming, orthogonal, and unitary). Shown are the cross-entropy loss and accuracy during training on a copy task with delay length $T_{\text{delay}} = 40$, evaluated across four independent random seeds.

full Fisher-information flow of an LSTM or GRU, and we do not claim any architectural advantages or theoretical optimality for these models.

However, an important observation is that *each gate* in an LSTM computes a recurrent transformation of the form

$$\text{gate}_t = \sigma(Wx_t + Uh_{t-1} + b), \tag{27}$$

which is structurally identical to the RNN update analyzed in our framework. Thus, while the theory does not extend to the entire LSTM architecture, it is still meaningful to ask whether initializing these gate-level recurrent components using Fisher–optimal weights improves training performance on tasks requiring long-range memory.

To test this, we replaced the RNN in the copy task with an LSTM and initialized the recurrent matrices of each gate using our Fisher–optimal rule, comparing against standard schemes (Xavier, Kaiming, orthogonal, and unitary). As shown in Fig. 15, Fisher–optimized initialization consistently ranks among the top-performing schemes across training runs.

These results provide empirical evidence that—even though our theory is developed strictly for RNNs—the Fisher–optimal initialization of the *RNN-like components* inside an LSTM can still enhance temporal sensitivity, reduce gradient degradation, and improve learning efficiency on long-distance sequential-memory tasks. Importantly, this observation supports the generality of the information-flow perspective, while remaining consistent with the limitations of our theoretical framework.

### A.6.3. SEQUENTIAL MNIST

Following the main text, we include additional experiments testing the training performance of networks initialized with Fisher–information–optimized weights versus standard schemes (Xavier, Kaiming, orthogonal, and unitary). Because the recurrent weights are fixed in this setup, the task provides a direct assessment of how different initializations affect the network's ability to preserve information from inputs far in the past. In turn, this tests how well the preserved memory supports the effective training of the decoder (Fig. 16).

Across four random seeds, Fisher–information–optimized initialization yields consistently lower loss, higher accuracy, and faster convergence than all baseline initializations. These results confirm that Fisher-derived initializations improve

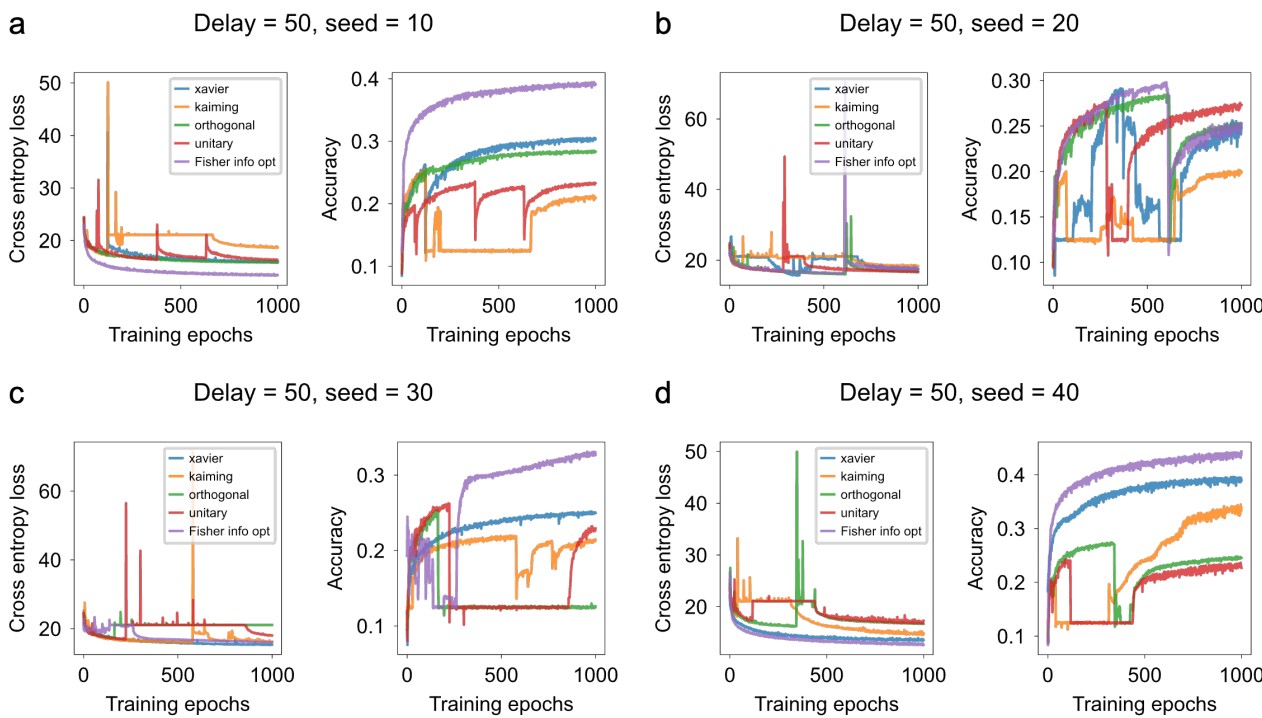

*Figure 13.* **Copy Task with RNN.** Comparison of Fisher–information–optimized initialization with standard initialization schemes (Xavier, Kaiming, orthogonal, and unitary). Shown are the cross-entropy loss and accuracy during training on a copy task with delay length $T_{\text{delay}} = 50$, evaluated across four independent random seeds.

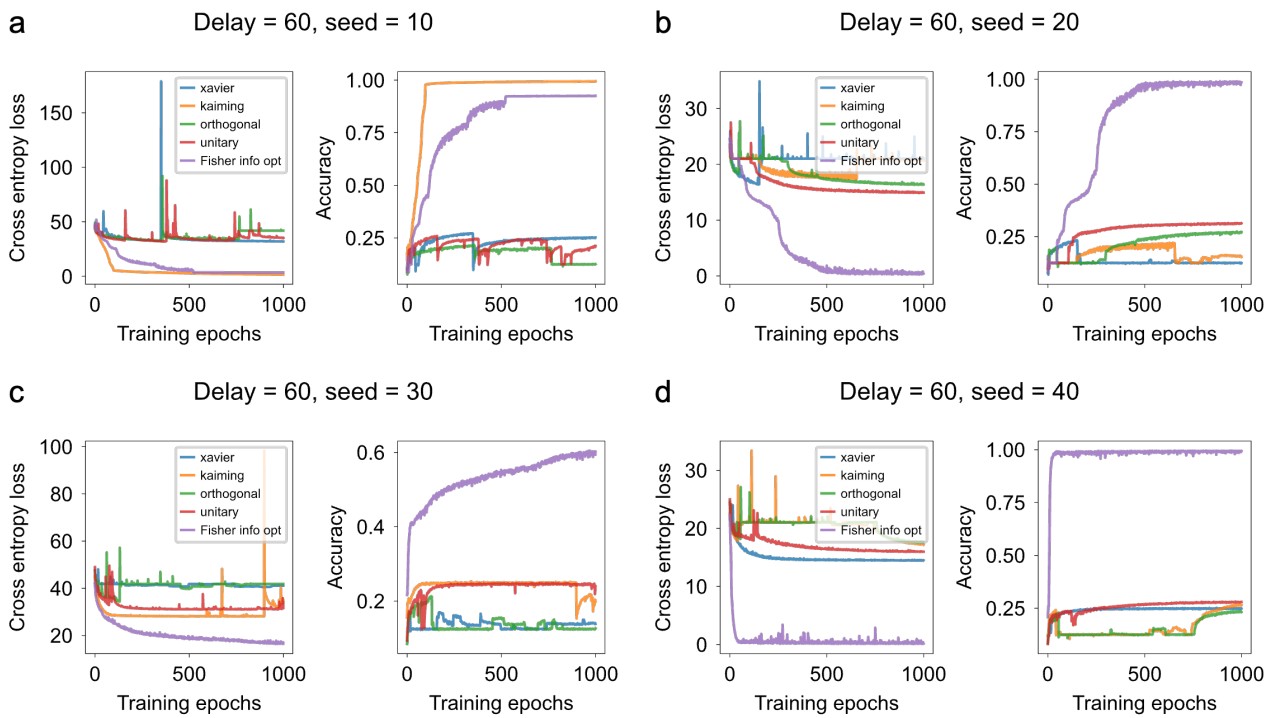

*Figure 14.* **Copy Task with RNN.** Comparison of Fisher–information–optimized initialization with standard initialization schemes (Xavier, Kaiming, orthogonal, and unitary). Shown are the cross-entropy loss and accuracy during training on a copy task with delay length $T_{\text{delay}} = 60$, evaluated across four independent random seeds.

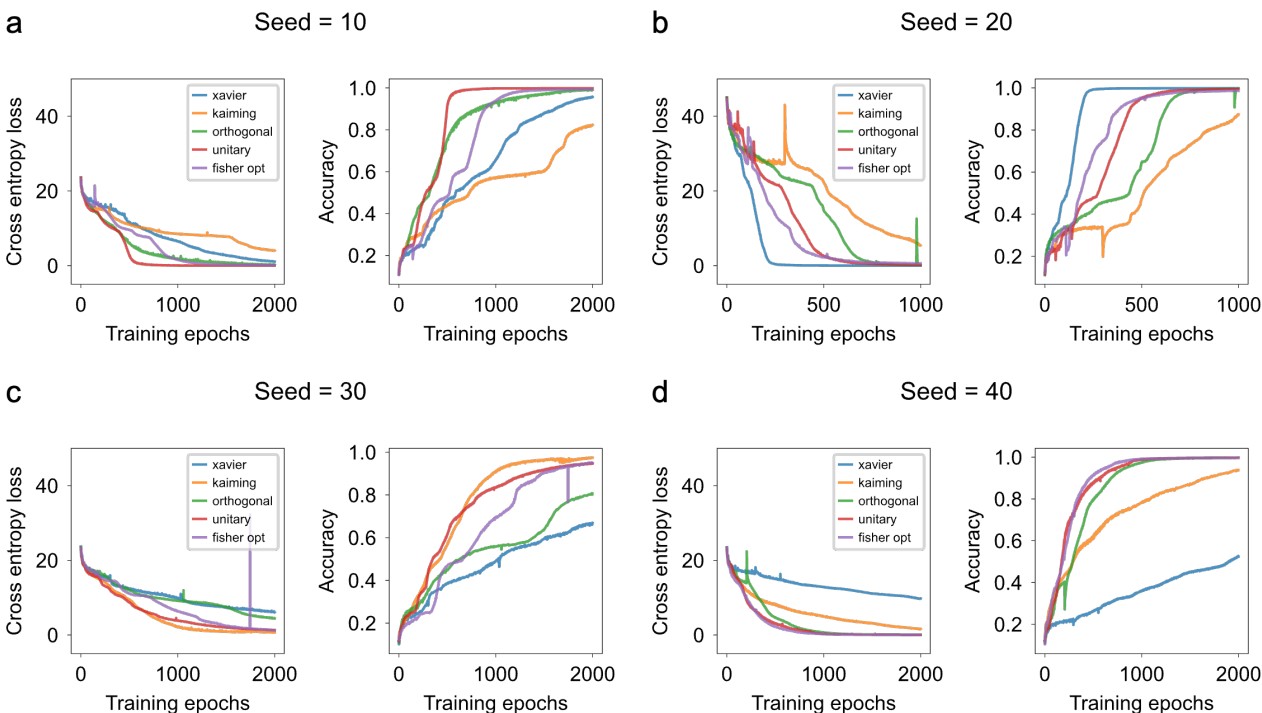

*Figure 15.* **Copy task with application of Fisher–optimized initialization to LSTM gates.** Each gate of the LSTM is initialized using the Fisher–information–optimized weights derived for recurrent networks. Although our theoretical framework formally applies to single–state RNNs, and thus does not fully capture the dual–state dynamics of LSTMs, the Fisher–based initialization still provides stable gradient propagation in practice. Shown are the cross-entropy loss and accuracy on the copy task with delay length $T_{\text{delay}} = 50$, compared with standard initialization schemes (Xavier, Kaiming, orthogonal, and unitary) across four independent random seeds.

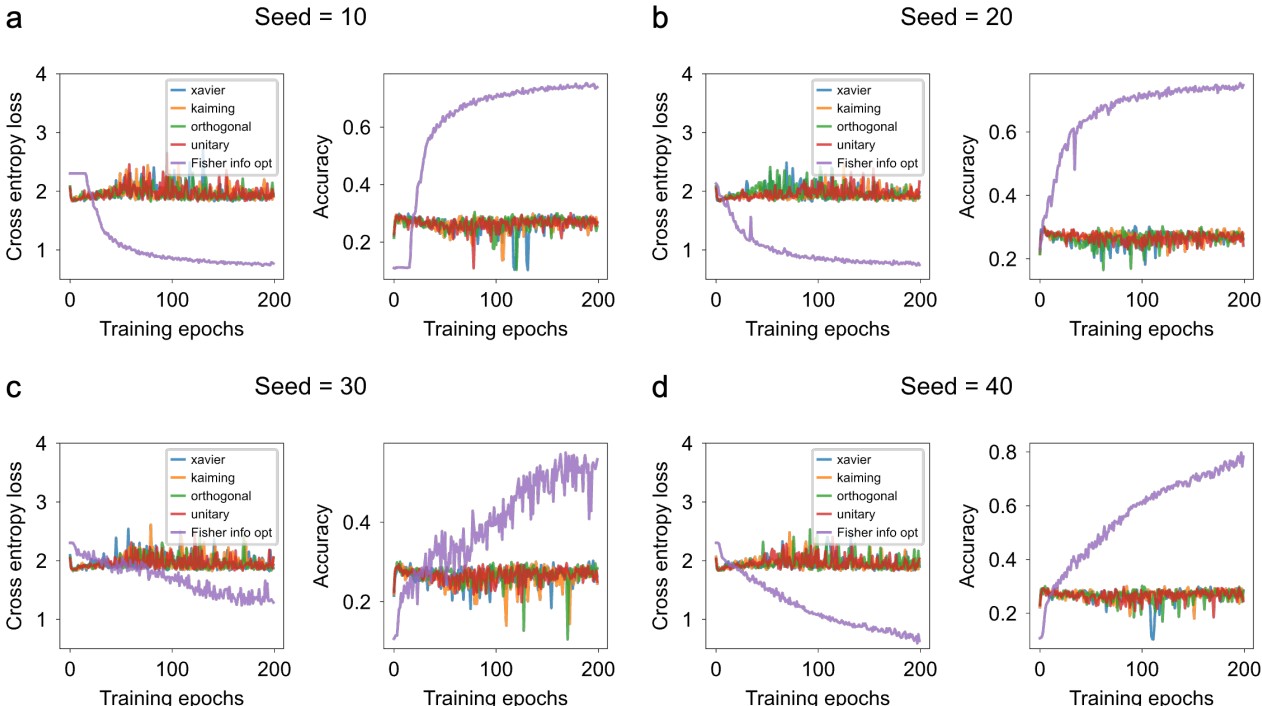

*Figure 16.* **Sequential MNIST with fixed RNN weights.** Training performance of networks initialized with Fisher–information–optimized weights compared against standard schemes (Xavier, Kaiming, orthogonal, and unitary). Plotted are the cross-entropy loss and accuracy on the Sequential MNIST task, evaluated across four independent random seeds.

long-range memory by keeping the neural activity at later time steps sensitive to perturbations in early inputs, thereby stabilizing gradient flow and facilitating more effective decoder training.

**The theory is general across input distributions.** Our theoretical results show that Fisher–information–optimal initialization preserves pairwise distances between inputs under the mean-field limit. Because these initialization rules are derived analytically—rather than learned from any specific dataset—the theory predicts that they will preserve input perturbations for *any* stimulus distribution, provided the network is sufficiently large relative to the sparsity of the inputs. This requirement directly parallels the conditions for approximate isometry in compressed-sensing theory and the Johnson–Lindenstrauss lemma (Foucart & Rauhut, 2013), where high-dimensional random projections preserve pairwise distances with high probability.

In Fig. 10, we show that the Fisher-information dynamics across subnetworks remain consistent across two qualitatively different datasets (CIFAR-10 and IndoorCVPR), regardless of whether the network is exactly at the Fisher-optimal point. This demonstrates that the structure of the initialization—and the resulting information flow—is governed by the theoretical framework rather than by dataset-specific statistics. The predicted population-level information dynamics therefore generalize naturally across domains.

To further evaluate this generality in a behavioral task, we conducted sequential-classification experiments on both CIFAR-10 and IndoorCVPR. CIFAR-10 has similar spatial resolution to MNIST ($32 \times 32$ vs. $28 \times 28$) with ten classes. IndoorCVPR is substantially more challenging, containing high-resolution images ($128 \times 128$) spanning 67 categories. Learning all 67 classes would require a substantially more expressive decoder, which is outside the scope of this work: our focus is on the encoding and memory dynamics of the recurrent network, not on optimizing a deep classification head. Therefore, to maintain the same simple architecture used throughout the paper—an RNN paired with a single-layer MLP decoder—we restricted IndoorCVPR to five representative classes. This setup ensures that differences in performance primarily reflect the network's memory properties rather than decoder complexity.

Across both datasets (Fig. 17 18), Fisher–information–optimized initialization consistently yields lower loss, higher accuracy,

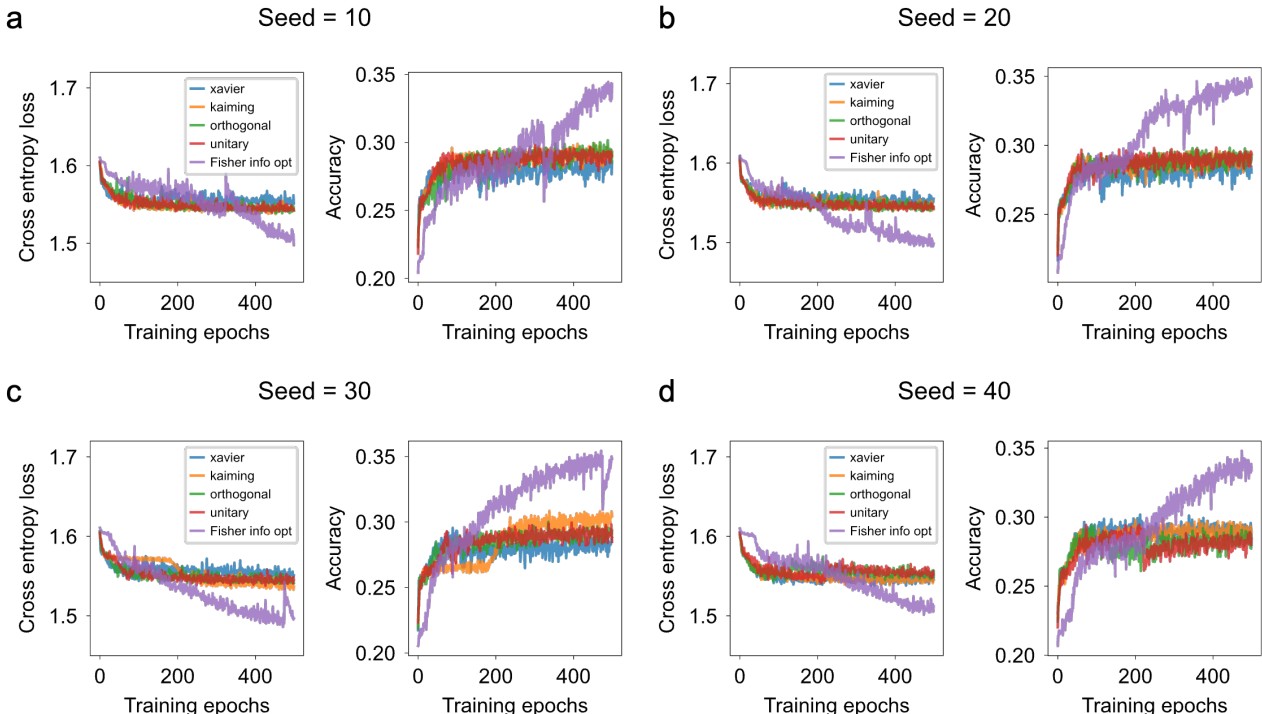

*Figure 17.* **Sequential classification on CIFAR10 with fixed RNN weights.** Using the same Sequential-MNIST framework, we apply the RNN to the CIFAR10 dataset. Training performance is shown for networks initialized with Fisher–information–optimized weights compared against standard schemes (Xavier, Kaiming, orthogonal, and unitary). Plotted are the cross-entropy loss and accuracy on the sequential classification task, evaluated across four independent random seeds.

and faster convergence than standard initialization schemes (Xavier, Kaiming, orthogonal, and unitary). These results mirror those obtained on Sequential MNIST and strongly support the theoretical prediction: Fisher-optimal initialization enhances long-range memory by maintaining sensitivity of the recurrent activity at late time steps to small perturbations in early inputs. This stabilizes gradient flow and allows the decoder to train more reliably and efficiently on sequential tasks.

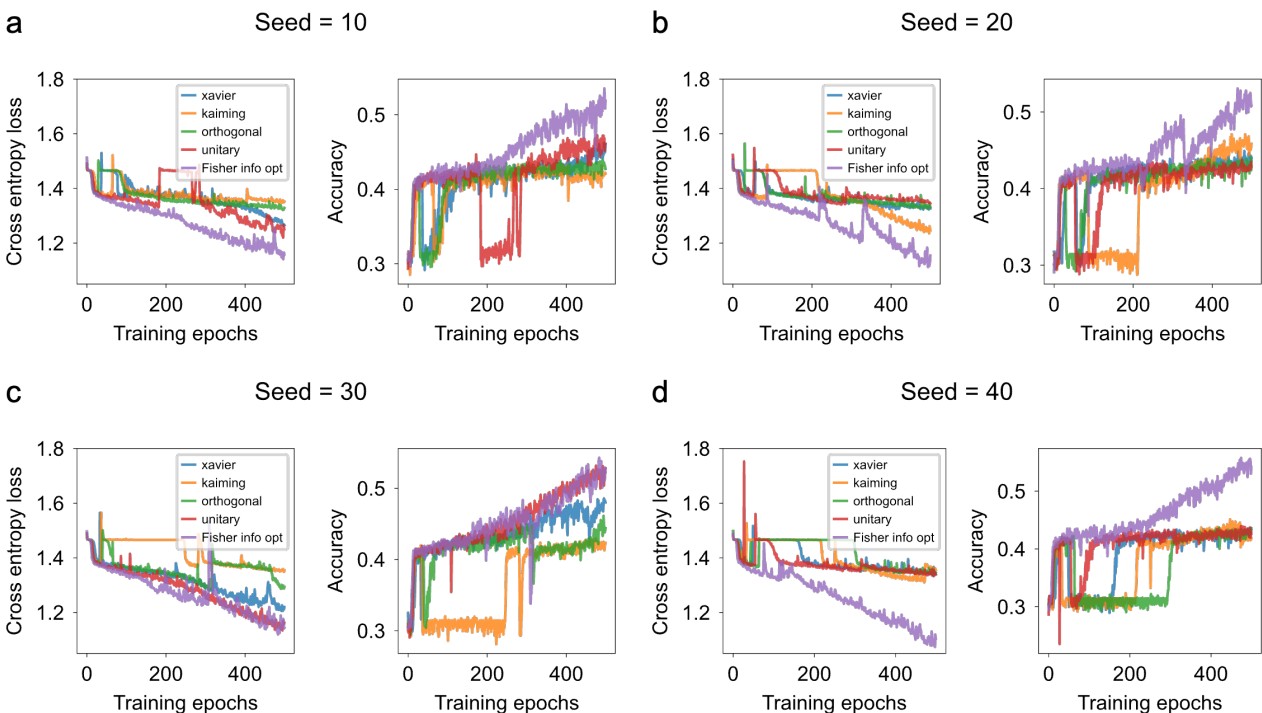

*Figure 18.* **Sequential classification on IndoorCVPR with fixed RNN weights.** Using the same Sequential-MNIST framework, we apply the RNN to the IndoorCVPR dataset restricted to five classes (airport_inside, artstudio, auditorium, bakery, bar). Training performance is shown for networks initialized with Fisher–information–optimized weights compared against standard schemes (Xavier, Kaiming, orthogonal, and unitary). Plotted are the cross-entropy loss and accuracy on the sequential classification task, evaluated across four independent random seeds.

