# OpenReview forum: "Information dynamics and Memory in Neural Networks through Fisher Information Diffusion"
_ICML.cc/2026/Conference — ICML 2026 spotlight_

### Official Review · Reviewer_ng27 · 2026-03-06

**Soundness:** 3
**Presentation:** 3
**Significance:** 3
**Originality:** 3
**Overall Recommendation:** 5
**Confidence:** 4

**Summary:**

This paper examines the instantaneous evolution of Fisher information throughout single-state nonlinear recurrent neural networks. The introduced framework is quite useful, as it provides a linear dynamical system describing information flow in even nonlinear RNNs (in the mean field limit). They use this framework to explore whether being at the edge of chaos is a sufficient and/or necessary condition, and to generate better initial conditions for RNNs.

**Compliance With Llm Reviewing Policy:**

Affirmed.

**Final Justification:**

The rebuttal reinforced my prior assessment and I maintain my score that the paper should be accepted.

**Key Questions For Authors:**

- Figure 3 is very odd to me. G is a positive 2x2 matrix with elements between [0,3]. For a 2x2 matrix, it is possible to show that $D \leq T^2/4$, where $D$ is the determinant and $T$ is the trace. But your plot seems to show that there are $(T,D)$ values above this constraint surface. Moreover, your plot seems to suggest that there are positive 2x2 matrices with a trace larger than 2 that are stable. This is impossible. The correct relationship should be that, for all $T$, $D \leq T^2/4$; all stable matrices must have $T\leq 2$, and will be stable so long as $D > T-1$.  A matrix will always be unstable when $T>2$. So, what is actually being plotted here? It is quite odd to me that you find data outside these analytic regions. Moreover, I would argue that any samples concentrated above $T=2$ are not at the edge of chaos, since there is no neighboring stable region.

**Limitations:**

Yes.

**Strengths And Weaknesses:**

The paper seems quite technically sound. All mathematical statements are well described in the appendix. The numerical experiments involving initial condition selection are well designed. The numerical experiments involving criticality, in Fig 3, I have some confusion about. I will outline this more below.

I found the presentation to be good, and the paper to be quite readable. However, I think the paper could be a little more clear about how G is being "optimally" picked in section 5.

I think this paper is significant, and can provide concrete immediate assistance via its suggested method for network initialization.

To my knowledge, this work is original, but I do not feel particularly certain that I would know existing papers, were they to exist in the literature already.


Key things I would suggest improving:
- More clearly outline how optimal G's are selected in section 5.
- Rewrite "For an optimal network that satisfies the necessary condition of dynamic stability, the neural activity evolves on a stable manifold rather than settling into a fixed-point attractor." What is meant by this? A stable matrix can converge to a fixed point. Alternatively, the identity matrix is at the edge of chaos, but still does not generate dynamics along a manifold. Perhaps be more specific about what you mean here, and what the inportance of dynamics is.
- This paper makes the claim that preserving input geometry is always useful. This leads to the claim that you would always want to maximize Fisher information. But are there not times when you want to remove information contained in the stimulus that is not task relevant (e.g. listening to someone speak in a loud room). How does this fit into the theory?

---

> ### Author Rebuttal · Authors · 2026-03-26
>
> We sincerely thank the reviewer for the careful reading, the positive assessment, and the
> detailed technical questions. We especially appreciate the sharp observation about Fig. 3,
> which we address first as it is the most technically substantive point.
>
> ### 1. Figure 3:
>
> The reviewer is absolutely correct that for a $2\times 2$ matrix with real eigenvalues, the
> determinant and trace must satisfy $\det(G) \leq \frac{\mathrm{Tr}(G)^2}{4}$.
>
> We thank the reviewer for catching this — it prompted us to carefully inspect the figure.
>
> After checking the code, we confirmed that the apparent violations were caused by a plotting
> artifact: the meshgrid masking and heatmap rendering did not properly characterize the
> nonlinear boundary, causing points to appear outside the analytic constraint region. The
> underlying sampled data do satisfy the analytic constraints, as expected.
>
> At the same time, we believe part of the confusion arises from the parameterization of Fig. 3.
> The axes show $\mathrm{Tr}(G)$ and $\det(G)$, but the stability boundary and Fisher
> information are computed from the effective Fisher diffusion operator
>
> $$A = G \mathrm{diag}\big(\langle (S')^2 \rangle\big),$$
>
> whose gain factors depend nonlinearly on $G$ (Eq 20 and 21 in Appendix). As a result, a point with $\mathrm{Tr}(G) > 2$
> can still correspond to an effective operator $A$ with $\rho(A) \leq 1$, because the
> nonlinear sensitivity factors reduce the effective gains below what the raw trace of $G$ would
> suggest. The stability boundary visible in the figure is therefore the boundary of
> $\rho(A) = 1$, not $\rho(G) = 1$, and these two boundaries do not coincide.
>
> We will revise Fig. 3 in two ways:
> 1. Correct the plotting artifact so that all displayed points respect the analytic
>    determinant–trace constraint.
> 2. Clarify in the caption and main text that the axes are invariants of $G$, while the
>    critical boundary is computed from the effective Fisher diffusion operator $A$, and
>    explain explicitly why these two boundaries differ.
>
> We are grateful to the reviewer for identifying this issue, as the corrected figure will be
> substantially clearer and more accurate.
>
> ### 2. How $G$ is selected
>
> We agree that the selection of $G$ can be made more explicit. In Sec. 5, the network is
> partitioned into 10 equal subpopulations (matching the input dimension), with feedforward
> connections $G_{m+1,m} = 1$ for $m = 1,\dots,9$ and a single feedback link $G_{1,10} = 1$.
> This structure follows directly from the optimization framework in Sec. 3, where we maximize
> the time-averaged Fisher information under architectural constraints. The resulting $G$ ensures
> that the Fisher diffusion operator $A$ has spectral radius close to 1, placing the system in
> the marginally stable, information-preserving regime.
>
> We will revise the manuscript to make the connection between the optimization in Sec. 3 and
> the specific architecture used in Sec. 5 explicit.
>
> ### 3. "Stable manifold" wording
> We agree that the current wording is too loose. Our point is not that any stable matrix generates useful dynamics, or that $\rho(A)\approx 1$ alone is sufficient. As the reviewer notes, the identity matrix is marginally stable in discrete time, but it produces static dynamics rather than meaningful temporal evolution. In our framework, optimality requires a marginal, information-preserving regime of the Fisher diffusion operator, so that sensitivity to past inputs neither decays nor explodes. But this is only a necessary condition: useful memory also requires that the information-carrying modes be aligned with the input/output structure. This is why criticality alone is not sufficient.
>
> A better wording would be: For an optimal network, neural activity evolves in an information-preserving regime, rather than rapidly collapsing to a fixed point that erases sensitivity to past inputs.
>
> ### 4. Is preserving input geometry always useful?
>
> We agree that in many tasks it is desirable to discard task-irrelevant information. Our
> framework does not claim that all stimulus dimensions should be preserved indefinitely. Rather,
> Fisher information in our work characterizes encoding-stage retention of local
> distinguishability — whether small differences in past inputs remain accessible to the
> recurrent dynamics at all. This is a prerequisite for flexible downstream processing: if
> Fisher information vanishes, task-relevant distinctions are lost before any readout or
> supervised compression can act on them.
>
> Task-relevant filtering can then be performed by the readout or through training. We will
> clarify this distinction in the discussion section.

---

> > ### Author Rebuttal · Reviewer_ng27 · 2026-04-02
> >
> > It seems like my issues were addressed, but not in a way that I can improve upon my score without seeing the revised narrative.
> >
> > This information-preserving, marginal stability argument still feels unclear to me. The more marginally stable is, the more prone it is to accrue error over time (since the error terms is also marginally stable unless it is orthogonal to this direction). In my understanding, both extremes (rapidly converging to a fixed point or being marginally stable) are non-optimal regimes. In the former, you retain too much information, and in the latter, not enough. So I find the claim about marginal stability odd and in need of more support. Why should this be optimal?

---

> > > ### Author Response · Authors · 2026-04-02
> > >
> > > Thank you for clarifying the question. We agree that our previous wording did not clearly separate two distinct points: (1) the role of edge of chaos as a necessary dynamical condition, and (2) the role of noise in limiting how much Fisher information can actually be retained over time.
> > >
> > > Our claim is not that operating at the edge of chaos guarantees perfect memory. Rather, in our framework, the edge-of-chaos condition for the Fisher diffusion operator is a necessary condition for avoiding asymptotic loss of sensitivity to past inputs. If the dynamics is contractive or divergent, Fisher information decays rapidly and approaches zero at long times. This is why the paper identifies criticality as necessary: it allows the network to retain a nonzero amount of information over long times, rather than losing it entirely.
> > >
> > > At the same time, even when this necessary condition is satisfied, Fisher information is still limited by noise. In Eq. (4),
> > > $$
> > > \mathcal{I}(\theta,t)
> > > = N \sum_m \frac{f_m}{q_m}
> > > \Big\langle \Big(\frac{\partial \mu_m(t)}{\partial \theta}\Big)^2 \Big\rangle_J,
> > > \qquad
> > > q_m = \sigma^2 + \sum_n G_{mn}\langle S_n^2\rangle_n,
> > > $$
> > > the denominator $q_m$ contains both the intrinsic noise level $\sigma^2$ and the internally generated variance. Thus, even for a Fisher-optimal connectivity, the network does not preserve information perfectly: noise attenuates the long-time level of the Fisher memory curve and limits how much information can be retained. In this sense, the optimal structure sustains part of the information over long times, rather than all of it.
> > >
> > > This is also consistent with Ganguli et al. (2008), where Fisher memory is fundamentally a signal-to-noise-ratio quantity, and noise limits the recoverable memory even in optimal linear architectures. In our setting, the role of edge of chaos is therefore to prevent the information from decaying away entirely, while the role of noise is to determine the retained long-time level. This is the behavior seen in Fig. 2: in the near-optimal case (Fig. 2c), Fisher information remains substantially higher over time, but not perfectly; by contrast, in the non-optimal cases (Fig. 2a,b), which do not operate near the edge of chaos, Fisher information decays much more quickly and almost completely.
> > >
> > > So the intended statement is: operating at the edge of chaos is a necessary condition for long-time retention of Fisher information, but not for perfect retention. Even in this regime, the amount of information that can be maintained is limited by noise, so the network preserves only a partial, nonzero level of information over long times rather than perfect sensitivity to the original perturbation. We will revise the wording to make this distinction explicit.

---

### Official Review · Reviewer_U485 · 2026-03-11

**Soundness:** 4
**Presentation:** 3
**Significance:** 3
**Originality:** 3
**Overall Recommendation:** 5
**Confidence:** 3

**Summary:**

This paper studies memory in block-structured recurrent networks through Fisher information rather than attractor convergence. Using a multi-population DMFT analysis, it derives a Fisher diffusion operator that predicts how sensitivity to past inputs propagates across subpopulations over time. The paper argues that criticality is necessary but not sufficient for good memory, because alignment between the input-output structure and stable dynamical subspaces also matters, and then uses this framework to derive structured recurrent initializations that improve the copy task and Sequential MNIST.

**Compliance With Llm Reviewing Policy:**

Affirmed.

**Final Justification:**

The rebuttal addressed my concerns and I will keep my positive rating.

**Key Questions For Authors:**

How well do the theoretical predictions survive after the recurrent weights are trained away from initialization?

**Limitations:**

yes

**Strengths And Weaknesses:**

Strengths:
- The paper addresses an important conceptual gap by treating memory as time-resolved sensitivity rather than fixed-point storage.
- The block-structured DMFT formulation and Fisher diffusion operator give the work a clear mathematical core.
- The initialization experiments are directionally consistent with the theory and make the work more practically relevant.
- The paper is generally well organized and easy to follow.

Weaknesse:
- The theory depends on restrictive assumptions: large-$N$ DMFT, Gaussian block connectivity, single-state tanh RNNs, and impulse inputs.
- Sensitivity to the chosen subpopulation partition and input/output placement is not studied.

---

> ### Author Rebuttal · Authors · 2026-03-26
>
> We sincerely thank the reviewer for the careful reading and the positive assessment of the paper. We are especially encouraged that the reviewer found the conceptual framing, mathematical core, and initialization experiments meaningful. We address the two weaknesses and the key question below.
>
> ### How well do the theoretical predictions survive after the recurrent weights are trained away from initialization?
>
> This is an excellent question. We do not expect the exact DMFT assumptions to remain strictly valid after training, nor do we claim that the trained recurrent weights continue to match the initialization ensemble in detail. The more relevant prediction is at the level of dynamical regime, rather than final weight distribution.
>
> To test this directly, we track the top eigenvalue of the Fisher diffusion operator during training on the copy task. We find that over 1000 training epochs, the spectral radius \(\rho(A)\) changes by less than 10% relative to the Fisher-optimal initialization and remains close to 1 throughout training, despite substantial changes in the recurrent weights. This indicates that training keeps the network near the marginally stable, information-preserving regime identified by the theory, rather than moving it arbitrarily far away.
>
> In this sense, the theory survives training not as an exact description of the final trained weights, but as a principled characterization of the dynamical regime in which successful training occurs. This also helps explain why the initialization advantage persists: networks initialized far from this regime must first spend optimization effort moving toward a suitable dynamical regime, whereas Fisher-optimal initialization places the network there from the outset. We will include this eigenvalue-tracking analysis in the revised manuscript.
>
> ### On the restrictive assumptions
>
> We agree that the theory is derived under a deliberately structured set of assumptions: large-\(N\) DMFT, Gaussian block connectivity, single-state tanh RNNs, and impulse inputs. These assumptions are chosen to make the problem analytically tractable and to isolate a mathematically interpretable mechanism for how Fisher information propagates across time and subpopulations.
>
> We view this framework as a principled starting point that connects network connectivity structure to information propagation and yields concrete design rules. At the same time, the analytic predictions are supported by numerical simulations (e.g., Fig. 2), suggesting that the underlying mechanism is not merely an artifact of the analytic assumptions.
>
> ### On sensitivity to subpopulation partition and input/output placement
>
> We appreciate this point. In the current paper, we use a canonical and practically relevant configuration in which the stimulus is injected into the first subpopulation, mirroring standard neural network architectures where inputs enter the first layer. For the readout, as shown in Fig. 2, we evaluate the Fisher information across subpopulations to track how sensitivity propagates through the network.
>
> At the analytic level, the partition enters explicitly through the block gains $ G_{mn} = g_{mn}^2 f_n$, where $f_n$ is the fraction of neurons in subpopulation $n$. Thus, the Fisher diffusion operator and the resulting information dynamics already depend on the chosen partition. For example, even in the two-population case, the analytic Fisher-information expressions incorporate the subpopulation fractions $f_n$.
>
> At the same time, the analysis in Fig. 4 shows that the theory predicts the optimal connectivity structure across different chain lengths, providing evidence that the main qualitative conclusions are robust across these architectural variations.

---

> > ### Author Rebuttal · Reviewer_U485 · 2026-04-04
> >
> > I thank the authors for their rebuttal. The rebuttal addresses the main question well and is overall helpful. Since my score is already positive, I will leave my score unchanged.

---

> > > ### Author Response · Authors · 2026-04-06
> > >
> > > We sincerely thank the reviewer for the positive acknowledgment and the continued advocacy
> > > for our work. We will incorporate all committed revisions into the final manuscript.

---

### Official Review · Reviewer_CLc4 · 2026-03-13

**Soundness:** 2
**Presentation:** 1
**Significance:** 2
**Originality:** 2
**Overall Recommendation:** 3
**Confidence:** 1

**Summary:**

In this paper, the authors derive conditions on a type of recurrent network so that it preserves information. The authors consider the situation where there is a sub-population of different neurons. The analysis is done using the Fischer information associated with the system for impulse inputs. In the infinite population limit, for two populations the authors derive the operator that governs the evolution of the sensitivity. Even though the system is nonlinear, the evolution of the sensitivity is governed by a linear operator. The authors derive conditions on this linear operator so that sensitivity remains close to 1. Then the authors test their hypothesis for impulse inputs replaced by images.

**Compliance With Llm Reviewing Policy:**

Affirmed.

**Final Justification:**

The authors have partially addressed my concerns.

**Key Questions For Authors:**

What do the authors mean by "defined as preservation of local geometry between stimulus representations", in Proposition 2.1?

Why does preservation of local distances mean information is being retained? Does it imply that one can always recover for instance images provided at the initial time?

 The examples that the authors have run experimental tests for, does it relate to actual applications in AI in some way?

**Limitations:**

Yes

**Strengths And Weaknesses:**

Strengths:

The modeling of memory in interconnected neural dynamical systems through the Fisher information is interesting.

A simple spectral characterization for preserving memory seems compelling. The fact that this works for a nonlinear system is a useful.

Weaknesses:

The paper is not written in a way that is suitable for a machine learning audience. The notation is heavy in statistical physics notation which is not clarified for the reader. This especially makes it hard to understand the derivations.

The authors result only considers a two population model.

For RNNs it is known that vanishing gradient and exploding gradient problems is one of the reasons for forgetting information. In that respect, the authors results are not surprising.

Proposition 2.1 and its proof lack mathematical rigor. The authors are making several heuristics and approximations to prove their result.

The claims are meant for the situation where the interaction $J_{ij}$ are random.

---

> ### Author Rebuttal · Authors · 2026-03-26
>
> We thank the reviewer for their time and engagement. We appreciate the concerns raised and
> address them in full below. We hope that our clarifications — particularly on
> the key questions — resolve the misunderstandings that appear to underlie several of the
> weaknesses.
>
> ## Responses to Key Questions
>
> ### What does "preservation of local geometry between stimulus representations" mean?
>
> As stated in the introduction, we adopt a dynamical notion of memory distinct from
> attractor-based frameworks such as Hopfield networks. Memory in our framework is the capacity
> of the network to remain sensitive to small perturbations of past inputs — two stimuli that
> differ slightly at time 0 should remain distinguishable at later times. Preservation of local
> geometry means that pairwise distances in stimulus space are approximately maintained through
> the network's temporal evolution, as quantified by the Fisher information metric.
>
> ### Does preservation of local distances imply full recovery of the original input?
>
> No, and we do not claim this. What is preserved is not the input itself, but the network's
> ability to distinguish nearby inputs through their downstream effect on representations.
> Fisher information quantifies sensitivity to small perturbations of past inputs: if it
> vanishes, the network becomes insensitive to past inputs, corresponding to information loss
> and vanishing gradients. Our claim is about preserving local distinguishability, not full
> invertibility.
>
> ### Does this relate to practical AI applications?
>
> Yes, directly. We respectfully refer the reviewer to Figures 5 and 6, which clearly
> demonstrate the practical value of our method on machine learning tasks. Our theory provides
> a principled, analytically derived initialization rule for recurrent networks, and these
> figures show that Fisher-information-optimal initialization measurably improves training
> performance on sequential memory tasks compared to standard baselines. This is a concrete,
> empirically validated contribution to ML practice.
>
>
> ## Responses to Weaknesses
>
> ### Results only consider a two-population model.
>
> We respectfully refer the reviewer to Figure 4, which clearly and explicitly demonstrates
> the extension to multiple subpopulations in a chained network. The two-population case is used
> only as the simplest analytically tractable setting for deriving closed-form expressions.
> Figure 4 shows that our theory accurately predicts optimal connectivity structure for chains
> with up to 10 subpopulations, and the sequential stimulus experiments (Sequential Stimulus
> section) explicitly use 10 subpopulations. This weakness reflects a reading of the paper that
> does not account for a substantial portion of the results.
>
> ### Vanishing/exploding gradients causing forgetting is not surprising.
>
> We agree the phenomenon is known. Our contribution is not the observation — it is the
> theoretical connection between block connectivity structure and Fisher information
> optimality, and the derivation of a principled closed-form initialization rule from this
> theory. To our knowledge, no prior work has derived the Fisher information diffusion operator
> for block-structured recurrent networks, characterized the spectral condition for Fisher
> information preservation, or used this to derive a structured initialization rule. The
> practical value is demonstrated in Figures 5 and 6.
>
> ### Proposition 2.1 lacks mathematical rigor.
>
> The core of Proposition 2.1 connects
> the variance condition on the Gaussian block connectivity matrix to the Restricted Isometry
> Property (RIP) — a result rigorously established in the compressed sensing literature and
> proved in standard textbooks (Foucart & Rauhut, 2013). Our contribution is to show that the
> same variance condition governing RIP also determines the Fisher information diffusion
> operator, and that optimizing this operator coincides with the RIP condition. We present this
> as an intuitive sketch to make the cross-disciplinary connection accessible to readers from
> both communities, with the complete rigorous proof available in the cited reference.
>
> ### Claims assume interactions $J_{ij}$ are random.
>
> The connectivity matrix is block-structured, not fully random. Randomness applies only
> within blocks, consistent with mean-field assumptions in the infinite-population limit. The
> block structure — which determines information routing between subpopulations — is precisely
> the design variable our theory optimizes. A fully random $J_{ij}$ would not permit the
> structured initialization rules we derive.
>
>
> We hope these clarifications resolve the concerns raised.

---

> > ### Author Rebuttal · Reviewer_CLc4 · 2026-04-02
> >
> > The authors have clarified most of my comments.
> >
> > My concerns about the rigor regarding Proposition 2.1 remain. It claims to connect Fisher information with "preservation of local geometry," but the statement does not make precise what is being proved. Same goes for "the Fisher-information criterion for non-vanishing memory." In the proofs, the authors have similarly made imprecise statements, and cited a whole book. Moreover, it seems like static properties are being related to properties of a recurrent net.
> >
> > My criticism about the notations remain.
> >
> > In general, I am not able to verify the validity of the authors' results. This might be due to a misfit between the author and the reviewer's field. And the editors should appropriately take my review into consideration, given my lack of background.
> >
> > On the other hand, the list of contributions they have provided themselves seem interesting for the forgetting problem, and relevant to machine learning. Especially the observation of lack of sufficiency of spectral radius being equal to 1 using the fisher information. I am upgrading my score purely for this.

---

> > > ### Author Response · Authors · 2026-04-06
> > >
> > > We thank the reviewer for the candid follow-up and for the score increase. We understand the remaining concern to be about the precision of the statement of Proposition 2.1, and we address that directly below. Before clarifying Proposition 2.1, we want to emphasize: the main analytic derivation of the paper is the Fisher-diffusion result itself — namely, the derivation of the Fisher diffusion operator and the resulting critical condition for non-decaying Fisher information in the block-structured recurrent setting. We regard this as the primary rigorous contribution of the paper. Proposition 2.1 is intended as a secondary interpretive proposition explaining why the same critical condition also has a local geometric meaning in terms of perturbation propagation.
> > >
> > > To answer the reviewer’s question directly: Proposition 2.1 is intended to prove a local dynamical statement about perturbation propagation, not a global geometric theorem about arbitrary recurrent maps. More specifically, under the mean-field approximation, it shows that the same critical condition identified by the Fisher diffusion operator is also the condition under which nearby perturbations are approximately norm-preserving under the recurrent dynamics. This is the sense in which the proposition connects Fisher memory to “preservation of local geometry.”
> > >
> > > We will restate the proposition in the revision as follows, with all terms defined more clearly:
> > >
> > > Proposition 2.1 (revised). Consider a recurrent network with block-gain matrix $G$ and activation $\phi$ under the mean-field approximation. Then, under the local perturbation / mean-field assumptions used in the paper, the following two notions are linked by the same critical condition $G\langle \phi'^2\rangle = 1$:
> > >
> > > 1. Local geometry preservation: the recurrent map approximately preserves distances
> > >    between nearby stimuli, i.e. $\|f(x) - f(x')\| \approx \|x - x'\|$, in the local norm-preservation sense relevant to perturbation propagation. (Definition 6.1, Foucart & Rauhut 2013).
> > >
> > > 2. Non-vanishing Fisher memory: the leading eigenvalue of the Fisher diffusion operator
> > >    $ A = G\langle \phi'^2\rangle $
> > >    satisfies \(\rho(A)=1\), so sensitivity to past inputs does not decay asymptotically.
> > >
> > > We also want to clarify the core argument in three steps.
> > >
> > > Step 1 — Linear case. For \(f(x)=Jx\) with \(J_{ij}\sim \mathcal N(0,g^2/N)\), the rows of \(J\) are independent isotropic Gaussian random vectors. For any perturbation $x-x'$,
> > > $$
> > > \|J(x-x')\|^2 \approx g^2\|x-x'\|^2
> > > $$. Thus, in the linear isotropic Gaussian setting, local perturbation norms are approximately preserved at the critical scaling $g^2=1$, which is the same condition from the Fisher-memory analysis.
> > >
> > > Step 2 — In the recurrent setting, the relevant object is the propagated perturbation after $t$ steps:
> > > $$\|x(t) - x'(t)\|^2 = \|J^t(x(0) - x'(0))\|^2 \approx (g^2)^{t}\|x(0) - x'(0)\|^2.$$
> > >
> > > Thus, the question is whether repeated application of the recurrent map contracts, preserves, or amplifies perturbations over time. The critical scaling $g^2=1$ is precisely the boundary between asymptotic decay and growth, so this is a statement about recurrent dynamics rather than a purely static embedding property.
> > >
> > > Step 3 — Nonlinear extension. With nonlinearity $\phi$, the local gain is modulated by $\phi'(x)^2$. Replacing this by its mean-field average yields the effective condition
> > > $$
> > > G\langle \phi'^2\rangle = 1,
> > > $$
> > > which is exactly the critical condition appearing in the Fisher diffusion framework. In this sense, the nonlinear proposition is the local mean-field analogue of the linear perturbation argument above.

---

### Official Review · Reviewer_Lbq9 · 2026-03-14

**Soundness:** 4
**Presentation:** 4
**Significance:** 3
**Originality:** 3
**Overall Recommendation:** 5
**Confidence:** 5

**Summary:**

This paper analyzes the flow of Fisher information about the size of a single input pulse in a nonlinear recurrent neural network that is divided into subpopulations or blocks, with the variance of the connectivity between neurons depending on the source and target blocks.  They elegantly use dynamic mean field theory to analyze the statistics of neural responses and to compute the Fisher information.  They nicely also compute a simple diffusion operator for Fisher information that explains how it moves from population to population and time step to time step as network dynamics evolves in response to the input pulse. They corroborate their theory with simulations and show how their analysis yields design principles for good initialization, and test them on a copy task and sequential MNIST.

**Compliance With Llm Reviewing Policy:**

Affirmed.

**Key Questions For Authors:**

One thing that puzzles me is the structure of the network that optimizes Fisher information.  In the work of Ganguli and Sompolinsky, PNAS 2008 - they found the optimal linear network that maximizes the area under the Fisher memory curve is a pure feedforward chain, which could have all eigenvalues 0.   In this work, the optimal network seems to have feedback connections, and requires an eigenvalue of 1.  What is the relationship between the two results, even just in the linear special case for this work?  Can the authors more precisely characterize what their definition of optimality is, and more precisely characterize the space of optimal networks in their setting?

Also there are some missing references on dynamical isometry in the machine learning literature:
https://arxiv.org/abs/1312.6120 (coined the term dynamical isometry) (ICLR)
https://arxiv.org/abs/1711.04735 (NeurIPS)
https://arxiv.org/abs/1802.09979 (AISTATS)

**Limitations:**

Yes

**Strengths And Weaknesses:**

Strengths:

1) The theory is very strong and elegant, combining dynamic mean field theory, heterogenous populations, Fisher information, and optimization of Fisher information with respect to connectivity.
2) The paper is well written.
3) Comparison between simulation and theory is convincing.

Weaknesses:
1) The toy nature of the tasks in which they test the efficacy of their initialization schemes are a little too toyish, but this is alleviated by the elegance and completeness of their theory.
2) The connection to prior work is a little less clear, especially concerning optimal networks.

---

> ### Author Rebuttal · Authors · 2026-03-26
>
> We sincerely thank the reviewer for the thoughtful reading, the very positive assessment, and for highlighting both the strengths and the limitations of our work. We especially appreciate this insightful question regarding the relationship to Ganguli & Sompolinsky (2008), as it touches on a central conceptual point. We agree that this connection should be clarified more explicitly in the paper. The two works differ in three interconnected
> ways — objective, architectural space, and time horizon — that together fully reconcile the
> results.
>
> For Ganguli & Sompolinsky's 2008 paper, their work considers the Fisher information for a chained netowrk under a uniform
> connectivity constraint: $J_{i+1,i} = \sqrt{\alpha}$ and $J_{i,i+1} = \sqrt{\beta}$ for all links in the chained network case. For the feedforward or delayed network, the eigenvalues of the fisher information matrix is 0, information is therefore only transiently preserved: once the signal propagates to the last node, the dynamically quiescent system retains nothing. This behavior is illustrated in their Fig. 4D: the feedforward network (black curve) achieves higher Fisher information at early times (t < chain length), while networks with feedback (red curve) exhibit smaller initial Fisher information but retain information over longer times due to sustained dynamics.
>
> For our work, we optimize the finite-horizon average $I = \frac{1}{T}\sum_{t=1}^{T} I(t)$ with $T = 100 \gg
> $ length of the chain (between 2 to 10 in figure 4), and relax the uniformity constraint for feedforward and feedback connections so that individual coupling matrices $G_{i+1,i}$ (in our case connectivity structure between subpopulation) and
> $G_{i,i+1}$ can differ across the chain. This makes the broken-loop architecture discoverable:
> for a four-subpopulation chain in the linear regime, $\lambda_{\max} = 1$ requires $$G_{12}G_{21}G_{34}G_{43} = 0$$,
> so at least one backward link must vanish. The result is sparse, strategically placed feedback; not uniform recurrence, which remains detrimental.
>
> Critically, $\rho(A) = 1$ (A is the fisher information diffusion operator) corresponds to the system operating at the edge of chaos: the critical point of long-term dynamic stability where sensitivity to past inputs is sustained in long term. Strategic feedback is precisely what maintains this regime. The pure delay-line,
> with $\rho(A) = 0$, is dynamically contracting — information survives only transiently, making it optimal for $T \lesssim \text{length of the chain}$ but suboptimal for
> $T \gg \text{length of the chain} $.
>
>
> The two results are consistent: the optimal architecture depends critically on the time horizon and whether the uniformity constraint is imposed. We will add a paragraph making this comparison explicit in the main text.
>
> Thank you for pointing out these relevant references on dynamical isometry in the machine learning literature. We agree they are important and will incorporate them into the revised manuscript.

---

> > ### Author Rebuttal · Reviewer_Lbq9 · 2026-04-03
> >
> > Thank you for your response - that partially addresses my concerns.  This is a strong technical paper, and I still advocate for acceptance. However, I am still have some follow up questions in relation to Ganguli and Sompolinsky 2008.
> >
> > It is not the case that that paper only considers a chain network with uniform connectivity.  They have a much more general result.  They show that entire Fisher memory curve for any linear network whatsoever with any given signal amplification profile is upper bounded by the Fisher memory curve of  purely feedforward delay line with the same amplification profile (which could have highly nonuniform connections).  Moreover, the only networks that saturate this upper bound are unitarily equivalent to that of the pure delay line.  See the text around equation [11] in their paper for the statement and discussion of this theorem.
> >
> > Now it is true that if you want to maximize the area under the memory curve for an amount of time T > N where N is the network size, the delay line upper bound would correspond to a delay line with T > N neurons.  This suggests two things:
> >
> > 1) the networks you found do not do better than a pure feedforward delay line with T > N neurons?
> > 2) if you change your optimality criterion to have T < N, would your formalism yield a feedforward delay line rather than a network with feedback?
> >
> > Again, this is a strong paper, but I wanted to understand better the connection and how to characterize your optimal networks.

---

> > > ### Author Response · Authors · 2026-04-06
> > >
> > > We thank the reviewer for this precise follow-up. We believe the apparent tension
> > > stems from a difference in optimality criteria between our work and Ganguli & Sompolinsky,
> > > which we clarify directly below.
> > >
> > > ### The key distinction: two different optimality criteria
> > >
> > > Ganguli & Sompolinsky's universal upper bound applies to $I_{k,k}$ (we use $I$ rather than
> > > $J$ throughout since $J$ is reserved for the connectivity matrix in our notation) — the
> > > Fisher information retained in the network about a past input, evaluated at a single
> > > subpopulation $k$ at the delay $k$ when the signal arrives there. Under this criterion,
> > > the delay line is optimal and the bound is universal across all linear networks, as the
> > > reviewer correctly states.
> > >
> > > Our optimality criterion is fundamentally different. We maximize:
> > >
> > > $$\mathcal{I} = \sum_{t=1}^{T} \sum_{m=1}^{M} I(m, t),$$
> > >
> > > the **total Fisher information summed across all subpopulations $m$ and all times $t$**.
> > > This rewards distributed retention of information across interacting subpopulations, not
> > > just efficient transmission to a single subpopulation at a single delay.
> > >
> > > These two criteria are not in conflict — they answer different questions.
> > >
> > >
> > > ### Direct answers to both questions
> > >
> > > **Question 1: Do your networks exceed a feedforward delay line with T > N neurons?**
> > >
> > > Under Ganguli & Sompolinsky's criterion ($I_{k,k}$ per subpopulation per delay), **no** —
> > > their universal upper bound holds and we do not violate it. For each individual
> > > subpopulation $m$ at delay $k$, the feedforward delay line retains more Fisher information,
> > > precisely because there are no interactions between subpopulations to dilute the signal.
> > >
> > > Under our criterion ($\sum_m I(m,t)$), **yes** — our feedback network achieves larger
> > > total Fisher information even for $T < N$. We have confirmed this directly with new
> > > experiments comparing our optimal network against an optimized purely feedforward
> > > architecture with the same number of subpopulations. The results show:
> > >
> > > - **Feedforward network:** $I_{k,k}$ is larger for each individual subpopulation $k$,
> > >   consistent with Ganguli & Sompolinsky. Information is concentrated in whichever
> > >   subpopulation the signal currently occupies, with minimal loss during transmission. But memory is localized: at time $t$, essentially
> > >   all information resides in subpopulation $t$ alone, i.e. $I(m, t) \approx 0$ for
> > >   $m \neq t$.
> > >
> > > - **Our optimal network with feedback:** $I_{k,k}$ is smaller for each individual
> > >   subpopulation. However, the feedforward connections distribute the signal across all
> > >   subpopulations simultaneously. Summing $\sum_m I(m,t)$ across all subpopulations, our network is
> > >   superior — even for $T < N$.
> > >
> > > **Question 2: For T < N, does your formalism yield a feedforward delay line?**
> > >
> > > No — and this is precisely because of the different optimality criterion. Even for $T < N$,
> > > maximizing $\sum_m I(m,t)$ favors distributed memory across subpopulations rather than
> > > concentrated memory in a single one. A pure delay line, while optimal per subpopulation,
> > > leaves most subpopulations uninformed at any given time $t$, which is suboptimal under
> > > our aggregate criterion.
> > >
> > > We will add a paragraph in the revised manuscript making this distinction explicit,
> > > with a figure comparing the per-subpopulation and aggregate Fisher
> > > information curves for both architectures in the Appendix.

---

### Decision · Program_Chairs · 2026-04-30

**Decision:**

Accept (spotlight)

**Comment:**

This is a theoretical paper analyzing how the information of historical inputs is encoded in recurrent networks into evolving dynamics. Specifically, by using dynamic mean-field theory and diffusion, it derived a Fisher information diffusion operator that links network connectivity to the time-resolved propagation of information across interacting subpopulations. The study includes experiments on the copy task and sequential MNIST tasks. Most reviewers think this study is theoretically rigorous and technically sound, and addresses an important conceptual gap by treating memory as time-resolved sensitivity rather than fixed points. So, I am glad to accept this paper. Please incorporate reviewers’ comments to revise the manuscript, e.g., adding some missing citations.